# DomiRank Centrality reveals structural fragility of complex networks via node dominance

Marcus Engsig [1] ✉, Alejandro Tejedor [2,3,4] ✉, Yamir Moreno [2,3,5],
Efi Foufoula-Georgiou [4,6] & Chaouki Kasmi[1]

Determining the key elements of interconnected infrastructure and complex systems is paramount to ensure system functionality and integrity. This work quantifies the dominance of the networks' nodes in their respective neighborhoods, introducing a centrality metric, DomiRank, that integrates local and global topological information via a tunable parameter. We present an analytical formula and an efficient parallelizable algorithm for DomiRank centrality, making it applicable to massive networks. From the networks' structure and function perspective, nodes with high values of DomiRank highlight fragile neighborhoods whose integrity and functionality are highly dependent on those dominant nodes. Underscoring this relation between dominance and fragility, we show that DomiRank systematically outperforms other centrality metrics in generating targeted attacks that effectively compromise network structure and disrupt its functionality for synthetic and real-world topologies. Moreover, we show that DomiRank-based attacks inflict more enduring damage in the network, hindering its ability to rebound and, thus, impairing system resilience. DomiRank centrality capitalizes on the competition mechanism embedded in its definition to expose the fragility of networks, paving the way to design strategies to mitigate vulnerability and enhance the resilience of critical infrastructures.

Complex systems consist of many interacting components, with dynamics and emergent behavior being system properties. However, not all the constituents of such systems are equivalently central to their structure and dynamics, and in some systems, a few elements might be critical to ensure the integrity of the complex systems' structure or functionality[1–10]. Our capacity to accurately and efficiently identify key elements of such complex systems is at the core of actions as diverse as providing the most suitable website on an internet search[11], defining a vaccination scheme to mitigate the spreading of a disease[12–15], or

ensuring the integrity and functionality of transportation networks and critical infrastructures[16–20].

Network theory, by abstracting complex systems as a collection of nodes (system constituents) and links (interactions), has been instrumental in providing a general framework to assess different aspects of the relative importance of nodes in a network, yielding different node centrality definitions depending on the evaluated aspects, ranging from considering only the number of links a node has (degree centrality), aggregating the importance of

[1]Directed Energy Research Centre, Technology Innovation Institute, Abu Dhabi, UAE. [2]Institute for Biocomputation and Physics of Complex Systems (BIFI), Universidad de Zaragoza, Zaragoza, Spain. [3]Department of Theoretical Physics, University of Zaragoza, Zaragoza, Spain. [4]Department of Civil and Environmental Engineering, University of California Irvine, Irvine, CA, USA. [5]CENTAI Institute, Turin, Italy. [6]Department of Earth System Science, University of California Irvine, Irvine, CA, USA. ✉e-mail: marcus.w.engsig@gmail.com; alej.tejedor@gmail.com

a node's neighborhood (e.g., eigenvector[21], Katz[22], and PageRank[11] centralities) to considering the relative position of the node in the network (e.g., closeness and betweenness[23] centralities) or the role of the node in a dynamic process (e.g., current-flow[24], entanglement[25], and random-walk[26] centralities). The performance of these centralities is often benchmarked against each other in evaluating their capacity to generate targeted attacks to dismantle the network's structure or disrupt its functionality. In fact, centrality metrics have a pivotal role in designing mitigation strategies to enhance network robustness and resilience, critical emerging properties of utmost importance to maintain our day-to-day privileges and necessities, which heavily rely on interconnected infrastructures such as the internet[1,27,28] or the power grid[29–31].

In this work, we introduce a centrality metric called DomiRank centrality. Intuitively, it quantifies the degree of dominance of nodes in their respective neighborhoods. Thus, high scores of DomiRank centrality are associated with nodes surrounded by a large number of unimportant (e.g., typically low-degree) nodes, which they dominate. This new centrality gives importance to nodes based on how locally dominant they are, where the extent of the dominance effect can be modulated through a tuneable parameter ($\sigma$). Contrary to other centralities such as eigenvector or PageRank, and due to an implicit competition mechanism in the definition of DomiRank, connected nodes tend to have more disparate scores in terms of DomiRank centrality. We demonstrate that the inherent properties of DomiRank make both synthetic and real-world networks particularly fragile to the DomiRank centrality-based attacks, outperforming all other centrality-based attacks. Furthermore, we show that the DomiRank-based attack consistently outperforms most of the computationally feasible iterative (recomputed after each node removal) attack methods (e.g., degree, PageRank), and it causes more enduring damage than the efficient iterative betweenness attack. We provide both an analytical formula and a computationally efficient iterative algorithm for DomiRank, enabling it to be computed on graphical processing units (GPUs) with a parallelizable computational cost scaling with the number of links, allowing the centrality to be computed for massive sparse networks.

## Results

### Definition and interpretation of DomiRank Centrality

We define DomiRank centrality, denoted $\boldsymbol{\Gamma} \in \mathbb{R}_{N \times 1}$, as the stationary solution of the following dynamical process

$$\frac{d\boldsymbol{\Gamma}(t)}{dt} = \alpha A(\theta \boldsymbol{1}_{N \times 1} - \boldsymbol{\Gamma}(t)) - \beta \boldsymbol{\Gamma}(t), \tag{1}$$

where $A \in \mathbb{R}_{N \times N}$ is the adjacency matrix of the network $\mathcal{N}$ and $\{\alpha, \beta, \theta \in \mathbb{R}^+ : \lim_{t \to \infty} \boldsymbol{\Gamma}(t) = \boldsymbol{\Gamma} \in \mathbb{R}_{N \times 1}\}$. Note that the definition presented here is valid for unweighted, weighted, directed, and undirected networks, so in the more general case, a non-zero entry of the adjacency matrix $A_{ij} = w_{ij}$ represents the existence of a link from node $i$ to node $j$ with a weight $w_{ij}$. By expanding the term $\alpha A(\theta \boldsymbol{1}_{N \times 1} - \boldsymbol{\Gamma}(t))$, we obtain that the rate of change $\frac{d\boldsymbol{\Gamma}(t)}{dt}$ has a positive contributing term proportional to the nodal degree $\boldsymbol{k} = A \boldsymbol{1}_{N \times 1}$, and two negative contributing terms: the first proportional to the sum of $\boldsymbol{\Gamma}(t)$ over each node's neighbors ($A\boldsymbol{\Gamma}(t)$), and the second proportional to the current value of $\boldsymbol{\Gamma}(t)$. Thus, for the $i-$th node, Eq. (1) reads

$$\frac{d\Gamma_i(t)}{dt} = \alpha \left( \theta k_i - \sum_j w_{ij} \Gamma_j(t) \right) - \beta \Gamma_i(t). \tag{2}$$

For enhanced interpretability and without loss of generality, we discuss the case of an unweighted network, for which Eq. (2) reduces to

$$\frac{d\Gamma_i(t)}{dt} = \alpha \sum_{j \in \text{neighbors}_i} \left[ \theta - \Gamma_j(t) \right] - \beta \Gamma_i(t). \tag{3}$$

where neighbors$_i$ refers to the set of nodes directly connected to node $i$.

From a simple model perspective, $\boldsymbol{\Gamma}(t)$ can be interpreted as the evolving fitness of the individuals in a population subject to competition. Two different processes can alter the fitness of each individual: (i) Natural relaxation–fitness naturally converges to zero at a rate proportional to a constant $\beta$; (ii) Competition–individuals compete with each neighbor for a limited amount of resources, with their fitness reflecting their capacity to successfully maintain those resources. An individual's fitness tends to increase by being connected to neighbors whose fitness are below the threshold for domination ($\theta$) and decreases otherwise. Thus, the fitness of each individual changes proportionally to ($\sum_{j \in \text{neighbors}} \theta - \Gamma_j(t)$), where the proportionality constant is denoted by $\alpha$ and represents the degree of competition between neighboring individuals.

Notably, the fitness score of a given individual $k$ is a function of (i) its number of neighbors: the larger the number of neighbors of $k$, the more resources at stake, and therefore the larger the potential of $k$ to increase/decrease its fitness, and; (ii) its neighbors' neighborhood: having neighbors lacking dominance in their respective neighborhoods due to either the absence of neighbors or the presence of dominant neighbors increases the fitness of individual $k$. In other words, a given individual results in having a high value of fitness via the dominance of its neighborhood, either due to the direct dominance of its neighbors (quasi-solitary individuals) or via collusion (joint dominance) emerging from the synergetic action of several individuals in suppressing the fitness of a common neighbor while incrementing their respective fitness. The DomiRank centrality is thus based on the concept of dominance to provide scores to nodes that contextualize their importance in their neighborhood. Consequently, its direct interpretation in systems wherein interactions are mediated by dominance/power-based relations, such as Rich-Club networks[32–35], is apparent. In the Supporting Material (SM - see Section S-I), we provide illustrative examples of Rich-Club networks that shed more light on the relevance of the key factors controlling the emergence of dominant and dominated nodes via the joint dominance (collusion) mechanism and how to steer the relative power exploiting the concept of joint dominance.

From Eq. (1), we note that the centrality converges when $\alpha A(\theta \boldsymbol{1}_{N \times 1} - \boldsymbol{\Gamma}(t)) = \beta \boldsymbol{\Gamma}(t)$, for which the analytical expression (see Methods section for proof) of the DomiRank centrality $\boldsymbol{\Gamma} \in \mathbb{R}_{N \times 1}$ is given by:

$$\boldsymbol{\Gamma} = \theta \sigma (\sigma A + I_{N \times N})^{-1} A \boldsymbol{1}_{N \times 1}, \tag{4}$$

where $\{\sigma = \frac{\alpha}{\beta} \in \mathbb{R}^+ : \det(\sigma A + I_{N \times N}) \neq 0\}$. A convergence interval can be defined for $\sigma$, such that it is bounded as follows:

$$\sigma(\mathcal{N}) \in \left( 0, \frac{1}{-\lambda_N} \right), \tag{5}$$

where $\lambda_N$ represents the minimum (dominant negative) eigenvalue of $A$. Also note that the threshold for domination, $\theta$, only plays a rescaling role on the resulting DomiRank centrality, and therefore, we choose $\theta = 1$ without loss of generality.

We recall that the definition, interpretation, and use of DomiRank are valid for both undirected and directed networks. Only note that in the case of directed graphs, the adjacency matrix used in the definition of DomiRank (e.g., Eq. (4)) should correspond to the reverse of the

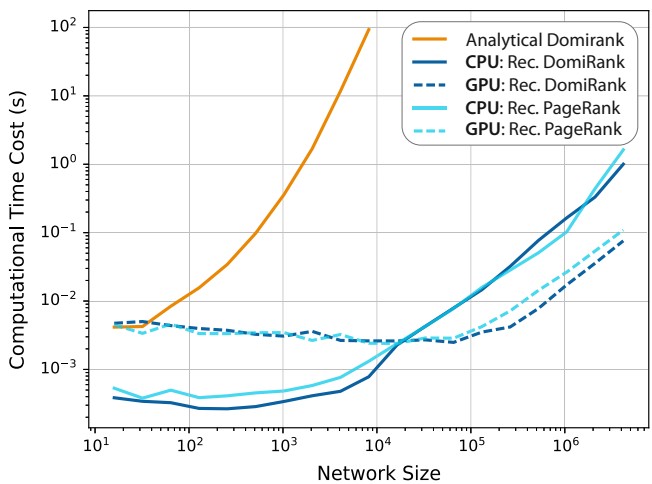

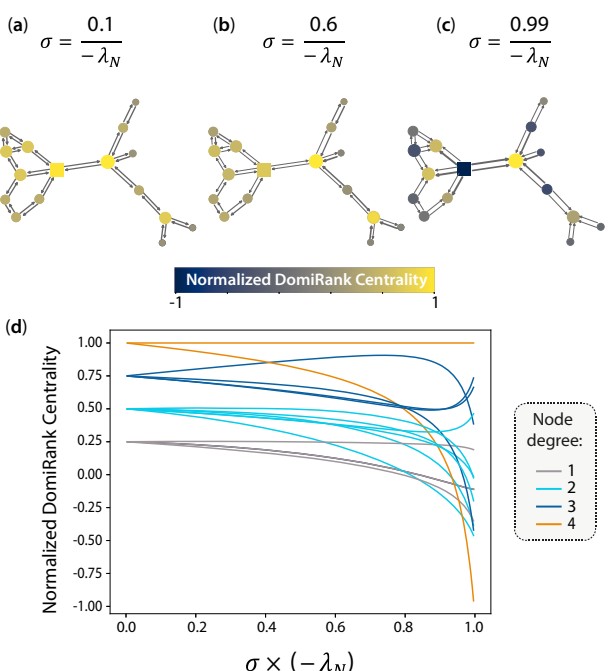

**Fig. 1 | Computational cost of DomiRank.** Mean (30 samples) computational costs to compute DomiRank analytically and estimate it recursively on a multi-threaded CPU and on the GPU as a function of the network size $N$. The mean DomiRank computational cost is also compared with the mean computational cost for estimating PageRank on the same multi-threaded CPU and GPU. The convergence criterion is evaluated using the $L1$ error between two consecutive iterations - i.e., $\frac{1}{N}||\mathbf{\Gamma}(t) - \mathbf{\Gamma}(t+dt)||_1 < dt \cdot \epsilon$, with a threshold set to $\epsilon = 10^{-6}$ (note that for the chosen convergence threshold, the Spearman correlation with the analytical solution is > 0.9999999).

graph relevant for the transfer of resources (e.g., information, traffic, etc.) to be consistent with the underlying concept of dominance.

## Numerical solution and computational cost of DomiRank

One of the key advantages of the proposed centrality is that it can be calculated efficiently through iteration in a parallelizable algorithm (see Fig. 1),

$$\mathbf{\Gamma}(t+dt) = \mathbf{\Gamma}(t) + \beta\big[\sigma A(\mathbf{1}_{N\times 1} - \mathbf{\Gamma}(t)) - \mathbf{\Gamma}(t)\big]dt, \qquad (6)$$

with a computational cost per iteration $C$:

$$C(A) = m + 5N, \qquad (7)$$

which scales with $\mathcal{O}(m+N)$, where $m$ is the number of links and $N$ is the number of nodes. Thus, the DomiRank scales with $\mathcal{O}(N^2)$ in the worst case (fully connected graph). Importantly, the algorithm can be distributed among $\kappa$ cores given that $\kappa \leq m$ for sparse matrices, which allows for parallel computation and efficient execution on GPUs. Figure 1 shows the computational costs of calculating DomiRank (analytically and recursively) and PageRank (recursively) for different network sizes, showing: (i) the high computational cost for the analytic computation of DomiRank for large networks, as it requires matrix inversion, (ii) the comparable computational cost of the recursive DomiRank to that of PageRank on both CPU and GPU infrastructure, and (iii) that the latency of computing recursive DomiRank on the GPU is the computational bottleneck unless the number of links is significantly larger the number of GPU cores, i.e., $m >> \kappa$. Thus, DomiRank centrality is computable even for massive (sparse) networks, allowing computational time costs under one second for networks consisting of millions of nodes.

## The role of DomiRank's parameter $\sigma$

The DomiRank centrality is modulated by the ratio $\sigma = \frac{\alpha}{\beta}$. To provide further insight into the effect of this parameter on the scores of the centrality, we explore DomiRank for varying values of $\sigma$ computed for a very simple network (see Fig. 2). As $\sigma \to 0$, the competition between the different nodes vanishes, and the

**Fig. 2 | DomiRank for different levels of competition ($\sigma$).** DomiRank centrality is displayed on the nodes of a simple network with $N = 15$ nodes for **a** low, **b** medium, and **c** large values of $\sigma$. Panel **d** shows the DomiRank centrality as a function of $\sigma$, wherein each solid line represents a specific node (color encoding node degree). In panels **a**–**c**, the direction of the pairwise transfer of fitness between nodes is shown by arrows, with their thickness representing the magnitude of that exchange. Note that for visualization purposes, the arrow thickness in panels **a**–**c** are scaled 25: 5: 1.

importance of the nodes reduces to their degree (see Fig. 2a, d and Eq. (4)). At the other end of the spectrum where $\sigma \to \frac{1}{-\lambda_N}$, the competition is maximum, and although the number of neighbors still plays a role, the network structure is the key feature defining the scores, where the synergistic competitive action of not directly connected nodes might result in their joint dominance in their respective neighborhoods. On that note, Fig. 2c,d shows how a node with a relatively high degree (square node) results in the lowest value of DomiRank centrality. This low value is the result of the joint domination by its four neighbors, which, despite having the same or lower degree as the dominated node, are able to increase their relative fitness by dominating their respective non-overlapping unfit neighborhoods and, together, the mentioned node. In fact, at the limit of high $\sigma$, each node tends to be either dominating its neighbor(s) or dominated by its neighbor(s) (see Fig. 2c). This effect is even more apparent when the direction of the steady-state pairwise contribution of the competition term to DomiRank is represented by arrows in Fig. 2c. Interestingly, in these extremely competitive environments, negative DomiRank scores emerge (see Fig. 2d). Individuals with deficit values of DomiRank are interpreted as fully submissive individuals who, instead of competing, directly give up their resources to neighboring nodes. This is the case for the node highlighted by its square shape in Fig. 2, which for highly competitive environments (Fig. 2c) experiences a reversal of its fitness exchanges (represented by the arrows) when compared with less competitive environments (Fig. 2a). Note that nodes exhibiting negative scores are able to maintain their steady-state DomiRank value due to the relaxation mechanism present in the model (see Eq. (1)). This mechanism effectively functions as a recovery or healing process for such nodes. Finally, intermediate values of $\sigma$ (e.g., see Fig. 2b) represent different domination strategies based on utilizing different

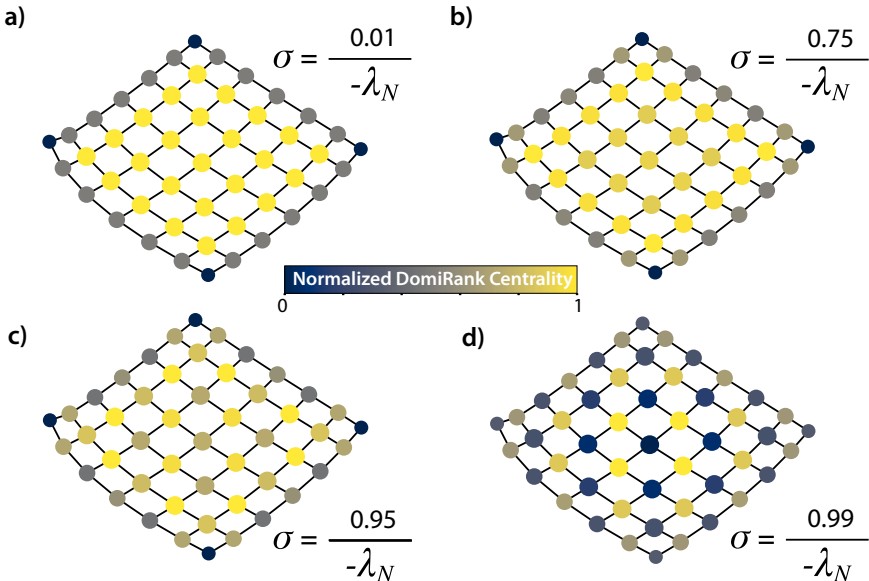

**Fig. 3 | The role of $\sigma$ in setting DomiRank values.** DomiRank centrality is displayed on the nodes of a 2D-square lattice with $N = 49$ nodes for different values of $\sigma$ **a** $\frac{0.01}{-\lambda_N}$, **b** $\frac{0.75}{-\lambda_N}$, **c** $\frac{0.95}{-\lambda_N}$ and **d** $\frac{0.99}{-\lambda_N}$ to illustrate different levels of competition, and how those levels set the trade-off between local (nodal) and global (meso- to large- scale structure) network information conveyed by DomiRank. Note that for each panel, the values of DomiRank are normalized to range in interval $[0, 1]$ for enhanced visualization.

balances of local node-based (low $\sigma$) and global network-structure-based (high $\sigma$) properties.

To develop some intuitive understanding of the role of $\sigma$ in setting up the trade-off between local (nodal) vs. global (meso- to large-scale) network properties in the resulting DomiRank distribution, we explore the DomiRank scores for different values of $\sigma$ in network with clear structure at the mesoscale level, a square lattice network (see Fig. 3). Note that we have chosen a small domain, $7 \times 7$ (49 nodes) to facilitate the visual interpretation of the results. When $\sigma$ approaches its lower bound (Fig. 3a), DomiRank converges to the node degree, relying solely on local information. As a result, all nodes except those on the lattice's edges tend to have nearly identical DomiRank values. As $\sigma$ increases, the DomiRank values begin to deviate from the node degree scores (Fig. 3b). This deviation occurs because each node's DomiRank becomes influenced by the values of their immediate neighbors. As a result, nodes directly linked to the lattice's edge nodes can partially dominate them, increasing their own DomiRank scores (note that this effect appears for significantly smaller values of $\sigma$ than the one displayed in Fig. 3b, but its visualization is less apparent using a consistent color scheme across panels). With further increments in $\sigma$ (Fig. 3c), the competition dynamics intensify. More internal nodes start sensing the lattice's boundary, and this influence propagates through the DomiRank scores, causing them to adapt based on dominance relationships. In essence, a node's DomiRank score is no longer solely determined by its immediate neighboring nodes; it also considers more distant features. Upon reaching the maximum $\sigma$ value (Fig. 3d), each node's DomiRank score is partially influenced by the entire network via the competition mechanism. For this extremely competitive setting, an ultimate alternating pattern of dominating and dominated nodes emerges shaped by two global network properties: the finite boundary and global symmetries. Thus, for example, in a square lattice with an even number of nodes, the pattern that emerges differs from the one shown in Fig. 3 due to the different constraints exerted by the system symmetry (for more details, see section S-II in the SM). Also, as an end member, a lattice with periodic boundary conditions, or an infinite lattice, all nodes are indistinguishable from any centrality metric perspective, including DomiRank.

## Dominance and network fragility

We also advocate for the capacity of DomiRank to reveal network fragility, both in terms of structure and dynamics. The rationale of this claim ties back to the two mentioned key factors dictating the DomiRank score of a node: its degree (number of neighbors) and the characteristics of its neighbors' neighborhoods (peripheries). In this context, nodes with high degrees that are connected to neighbors with few connections, i.e., sparse peripheries, are the prime candidates to achieve high DomiRank scores (being a star network, the end-member case of such a configuration). Those nodes are also central to network fragility, as their failure would lead to the fragmentation of their neighborhood. Interestingly, there exists an alternative source of heightened dominance that relies less on a node's local properties (degree) and more on its position within the global network structure. This kind of dominance primarily emerges in highly competitive environments characterized by high $\sigma$ values. Specifically, it results from joint dominance, where a group of nodes shares an overlapping neighborhood and lacks direct connections among themselves. Consequently, each of these nodes contributes to subduing dominance within the shared neighborhood. Notably, this mechanism also serves to identify vulnerable parts (structures) of the network, as the joint removal of the dominant nodes would lead to the fragmentation of their shared neighborhood, highlighting the fragility inherent in this structure.

Thus, DomiRank centrality-based attacks are anticipated to be very effective in dismantling networks because of their capacity to shatter the network in small components by targeting preferentially fragile neighborhoods. Also, from the point of view of the dynamics on networks, removing high-score DomiRank nodes could drastically disturb such dynamics, as those removals would appear as insuperable obstructions in sections of the system.

## Evaluating network robustness under DomiRank-based attacks

In order to gain further insight into the capabilities of DomiRank and to benchmark its performance with respect to other commonly used centralities, we examine the efficacy of targeted attacks based on DomiRank centrality for different network topologies, analyzing its

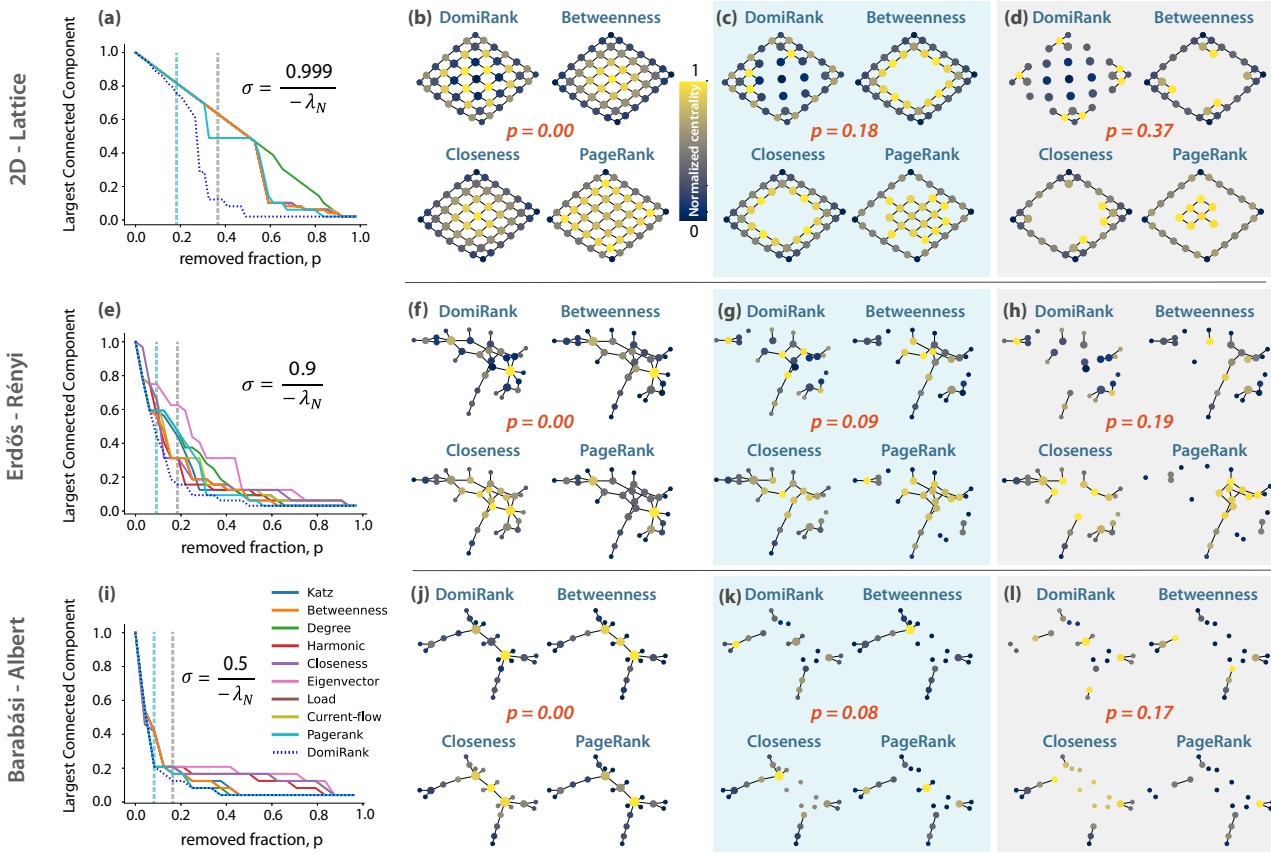

**Fig. 4 | Comparing DomiRank with other centralities in dismantling toy networks.** Evolution of the relative size of the largest connected component whilst undergoing sequential node removal according to descending scores of various centralities for three toy networks: **a** 2D regular lattice ($N = 49$), **e** Erdős-Rényi (ER; $N = 32$), and **i** Barabási-Albert (BA; $N = 25$). For each topology, panels **b**–**d**, **f**–**h**, and **j**–**l** show the graphical representation of the respective networks at various stages of the attack based on DomiRank, betweenness, closeness, and PageRank centralities. Note that the nodes are colored according to the relative value of the centralities normalized to be in an interval [0, 1] for enhancing comparability and visualization purposes.

ability to dismantle the network structure and functionality, and contrasting its performance with those of the attacks based on other centralities.

In this section, we first evaluate the structural robustness of different networks, both synthetic and real-world topologies, under sequential node removal (attacks) based on different centrality metrics and compare the results with those obtained based on DomiRank. To evaluate network robustness, we use its most commonly used proxy, the evolution of the relative size of the largest connected component ($LCC$)[36–39], whilst the network is undergoing sequential node removal. We compare the robustness of the different networks for the different attacks by directly comparing the resulting $LCC$ curves, and for simplicity and enhanced comparability, we also use the area under that curve as a summary indicator of robustness (the larger the area, the more robust the network is under that particular attack).

We start our analysis with synthetic toy networks, consisting of a reduced number of nodes, but wherein their graphical representation still allows us to visually identify patterns on the centrality distributions for different topologies, gaining insight into the interpretation of DomiRank and its performance when compared with different centralities. Particularly, we perform targeted attacks based on DomiRank and nine other centralities for three different topologies: 2D-regular lattice[40], Erdős-Rényi[41], and Barabási-Albert[42] networks. Note that for each topology, the range of $\sigma$ was explored to determine its optimal value to dismantle the network, i.e., minimize area under the $LCC$ curve (for other centralities such as Katz and PageRank, we used, without loss of generality of the results, the default values of 0.01 and 0.85, respectively—See SM section S-III for details). Figure 4a, e, i reveals that

the DomiRank centrality-based attacks dismantle these three networks more efficiently than all other tested centrality-based attacks. More particularly, DomiRank excels at dismantling regular networks (Fig. 4a). It is not surprising that for this topology, DomiRank centrality produces the most effective attack for large values of $\sigma$, wherein network structure is overweighed to the detriment of local node properties. This value of $\sigma$ leads to a DomiRank distribution wherein if a node is important (dominating node), all of its adjacent nodes are not important (dominated node), and vice-versa (Fig. 4b). An attack strategy based on such an alternating spatial pattern is significantly advantageous with respect to other traditional centrality-based attacks (Fig. 4a–d). This advantage stems from the strategic removal of existing neighbors, effectively isolating nodes and efficiently reducing the size of the largest connected component. Applying a similar DomiRank-based attack strategy to a more heterogeneous network, such as Erdős-Rényi (see Fig. 4e), still leads to the highest fragility of the network, also capitalizing on the built-in competition mechanism of DomiRank (high value of $\sigma$) that penalizes connections between nodes labeled as highly central, unless they possess disjoint neighborhoods to exert their respective dominance. For most of the other centrality metrics, including Betweeness, Eigenvector, PageRank, and Katz, highly central nodes permeate their centrality to their direct connections (see Fig. 4f–h). However, that by-contact importance only reflects the centrality of their truly important neighbor, yielding attack sequences less efficient than DomiRank. As networks display more hub-dominated topologies (e.g., scale-free), we expect that the optimal value of $\sigma$ for the most efficient attack decreases, emphasizing nodal properties (degree) with respect to the neighborhood structure. In the

toy example for a network generated by a Barabási-Albert model (see Fig. 4i), DomiRank still outperforms other centrality-based attacks in dismantling the network. In this case, the improvement is incremental since the most relevant information to destroy the network is local (node degree), and most of the centralities converge to a similar nodal ranking and attack sequence (Fig. 4j–l).

We further investigate the efficacy of the attack strategies based on the DomiRank centrality by dismantling larger synthetic networks ($N = 1000$) with varying degrees ($2 < \bar{k} < 20$) for numerous topologies. Particularly, we analyze the robustness of Watts-Strogatz[43], stochastic-block-model[44], Erdős-Rényi, random geometric graph[45], and Barabási-Albert networks, under ten different targeted attack strategies based on different centralities, including DomiRank, which revealed itself as the overall most efficient at dismantling synthetic networks (Fig. 5a–f). As hinted from our previous analysis of the toy networks, the margin by which the DomiRank-based attack outperforms the other strategies relates to the topological properties of the networks, which also dictate the optimal value of $\sigma$. Thus, for the Barabási-Albert topology (hub-dominated), DomiRank offers only an incremental improvement in the efficiency at dismantling the network (see Fig. 5f). On the other hand, for networks with meso-to-macro scale structural features (e.g., regularity) that dominate over the local node-based properties, DomiRank centrality significantly outperforms all other centralities (Fig. 5d). This also occurs for the Erdős-Rényi (Fig. 5b,e) and Watts-Strogatz networks (Fig. 5a). For a more detailed comparison between the different centralities, we refer the reader to the SM (section S-IV) where the correlation between them is displayed.

Real networks introduce several properties that are hard to produce simultaneously using generative models. Therefore for a more thorough and general benchmark of DomiRank, we analyze various real networks topologies of various sizes: (g) hub-dominated transport network (RyanAir connections)[46], (h) neural network (C-elegans)[3,47,48], (i) spatial network (power-grid of the Western States of the United States of America)[43], (j) citation network (high-energy-physics arXiv)[49,50], (k) massive social network (LiveJournal users and their connections)[50], and (l) massive spatial transport network (Full US roads)[49]. Our results are in line with those for synthetic networks, showing that the DomiRank is able to consistently dismantle the networks more efficiently than the other centrality-based attacks tested (see Fig. 5g–l). Another interesting phenomenon also observed for the synthetic networks is that the DomiRank-based attacks remove links more efficiently than previous methods (see Section S-VII in the SM). This means that for many of these networks, not only is the DomiRank better at reducing the size of the largest cluster size, but it also more severely cripples its connectivity, yielding not only to an overall faster but also a more thorough dismantling of the network. However, we note that for the social network analyzed (Fig. 5k), the PageRank-based attack outperforms the one based on DomiRank. We attribute this phenomenon to the presence of structural heterogeneity in the network topology (i.e., different structures in different subgraphs of the network). This heterogeneity hinders the assessment of node importance by DomiRank with a single value of $\sigma$ for the whole network. In the SM (section S-V), we provide evidence showing that, indeed, heterogeneity can lead DomiRank to underperform, hinting at potential approaches to address the evaluation of networks exhibiting heterogeneity.

As a last point of discussion about Fig. 5, we want to emphasize how the sensitivity of DomiRank to its unique parameter $\sigma$ becomes apparent from analyzing and displaying the results for such an extensive set of diverse networks jointly. This sensitivity, far from being a weakness, is a key strength of DomiRank, as it allows us to assess the important nodes in networks with topologies as different as a regular lattice and hub-dominated network. Thus, for any individual case, the edge that a DomiRank-based attack could offer over other centrality-based attacks could vary from being very significant (e.g.,

planar networks) to just marginal (e.g., scale-free). In fact, it is the consistent and sustained superior performance of DomiRank-based attacks across all ranges of topologies that makes these results particularly noteworthy overall.

The analysis of synthetic networks and real-world topologies has demonstrated the capacity of DomiRank to integrate local (node) and mesoscale information of the network, which, together with the competition mechanism embedded in its definition, produces centrality distributions that efficiently dismantle the networks by avoiding redundant scores in neighboring nodes (importance by-contact). This apparent handicap for other centralities could be addressed at the cost of recomputing the centrality distributions after each node removal. Note that this cost is prohibitive for distance-based or process-based metrics such as closeness, betweenness, or load centralities, even for networks of modest sizes as, for instance, betweenness has a computational complexity that scales with $\mathcal{O}(Nm)$ and $\mathcal{O}(Nm + N^2 \log N)$ for unweighted and weighted graphs respectively[51]. Despite this limitation, and for the sake of completeness, we also benchmark the DomiRank centrality distribution (computed once before the beginning of the attack) with the targeted attacks based on sequentially recomputed centralities. In this part of the analysis, we have incorporated attacks based on the Collective Influence (CI) algorithm[52–54], which is an iteratively recomputed centrality that aims to find the most influential nodes in a network. CI could be particularly relevant to our study as: (i) it can be mapped to an optimal percolation problem, and (ii) it has been successful in identifying previously neglected (non-locally important) nodes as important influencers.

Figure 6a–d displays the increase in performance of various centrality-based attacks when recomputed iteratively, particularly for betweenness centrality. In fact, for all the synthetic topologies tested, iterative betweenness and load centralities lead to the most efficient attacks at dismantling networks by a large margin. We also note, that CI is able to fully disintegrate the network (i.e., reduce all clusters to minimum size) the fastest, as expected from its mathematical formulation. However, the path of deterioration followed by the *LCC* during the attack is not among the most competitive in rapidly and sustainably reducing the size of the largest connected component. Notably, the attacks based on pre-computed DomiRank centrality generally outperform other attacks based on iterative centralities that are computationally feasible for medium, large, and massive networks - i.e., iterative degree, PageRank, eigenvector, Katz, and CI (except for ER and marginally for BA). Note that attacks based on iterative DomiRank centrality perform worse than the ones obtained from a single computation, which is actually expected as DomiRank leads to attack strategies aiming to cause structural damage, which requires the joint removal of several nodes. Therefore by recomputing DomiRank every time step, no coherent strategy emerges as the network structure becomes a moving target, i.e., the structure is re-evaluated at a faster rate (every node removal) than the time needed to remove the number of nodes necessary to inflict the structural damage. This fact underscores the intrinsic ability of the pre-computed DomiRank version to extract local information while considering the network's global context (with a larger value of $\sigma$ indicating a broader global context). In other words, DomiRank's underlying competition mechanism establishes an inherent, built-in process to prevent the assignment of artificial redundant scores to neighboring nodes. All the other displayed methods in Fig. 6a–d lack this intrinsic mechanism, which is compensated for through the sequential recomputation of the centralities after each node removal.

From all the attacks shown in Fig. 6, we want to highlight iterative betweenness, being the most efficient, and CI because it shares potential similarities with DomiRank. Attacks based on iterative betweenness centrality excel at destroying the *LCC* by finding *bottleneck* nodes instrumental in mediating most of the shortest paths and, thus, focusing on simply splitting the network. As a result of these

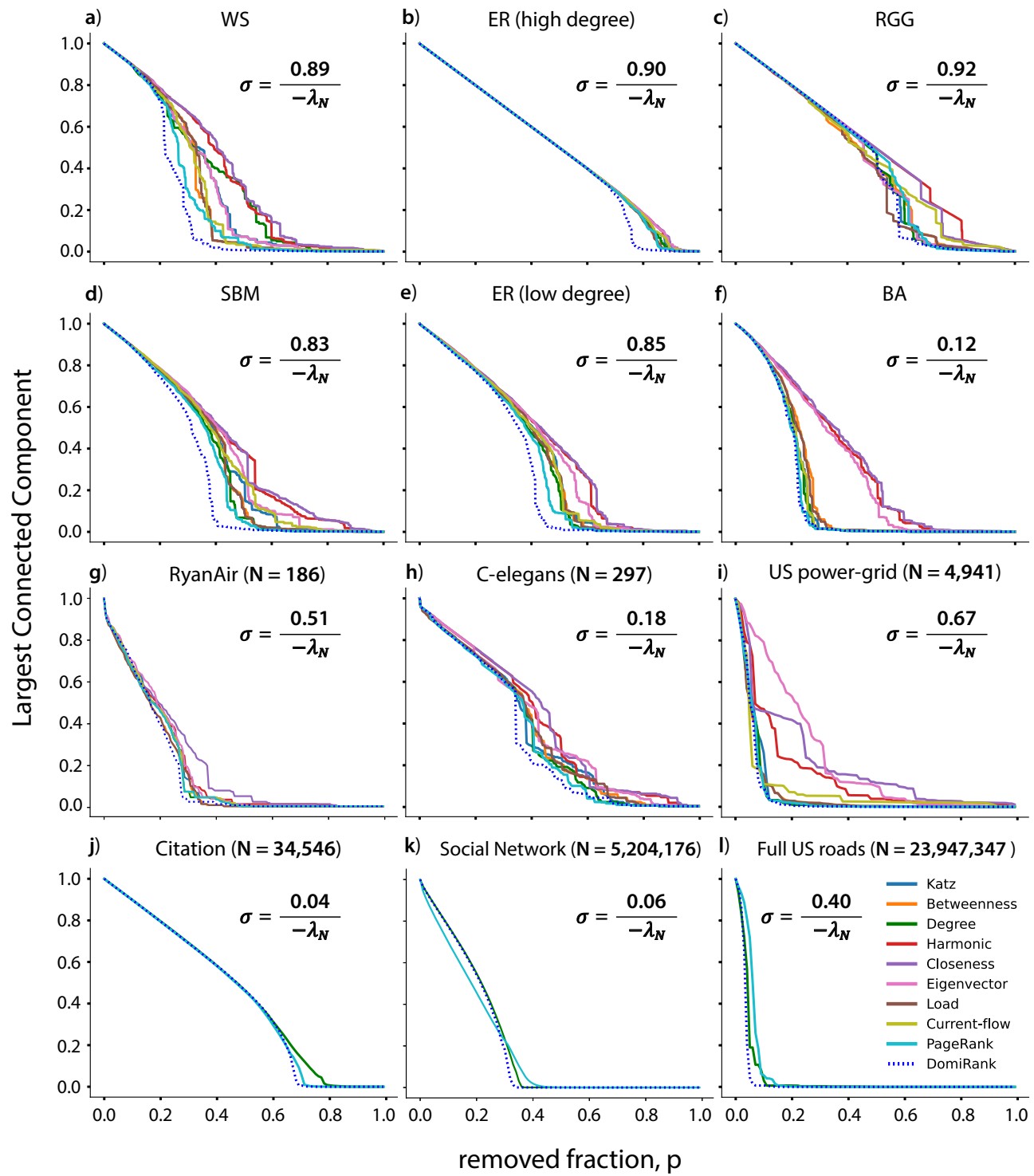

**Fig. 5 | Centrality-based attacks on synthetic and real-world networks.** Evolution of the relative size of the largest connected component (robustness) whilst undergoing sequential node removal according to descending scores of various centrality measures for different synthetic networks of size $N = 1000$: **a** Watts-Strogratz (WS; small-world, $\bar{k} = 4$), Erdős-Rényi (ER) with **b** high ($\bar{k} = 20$) and **e** low link density ($\bar{k} = 6$), **c** random geometric graph (RGG; $\bar{k} = 16$), **d** stochastic block model (SBM; $\bar{k} = 7$), and **f** Barabási-Albert (BA; $\bar{k} = 6$). The performance of the attacks based on the different centrality metrics is also shown for different real networks: **g** hub-dominated transport network (airline connections, $\bar{k} = 16$), **h** neural network (worm, $\bar{k} = 29$), **i** spatial network (power-grid, $\bar{k} = 3$), **j** citation network ($\bar{k} = 25$), **k** massive social network ($\bar{k} = 19$), and **l** massive spatial transport network (roads, $\bar{k} = 5$). Note that for panels **j**–**l**, where massive networks are shown, only a few attack strategies are displayed due to the impossibility of computation of the rest.

fundamental differences in the aim of the two centralities, we expect that despite the DomiRank-based attack being less efficient at dismantling the network than those based on iterative betweenness, it causes more severe and enduring damage, making it more difficult to recover from when compared with the damage produced by an iterative betweenness attack. CI-based attacks seek to find influential nodes according to their potential to cascade down information. As such, CI does not look only to local information but also integrates

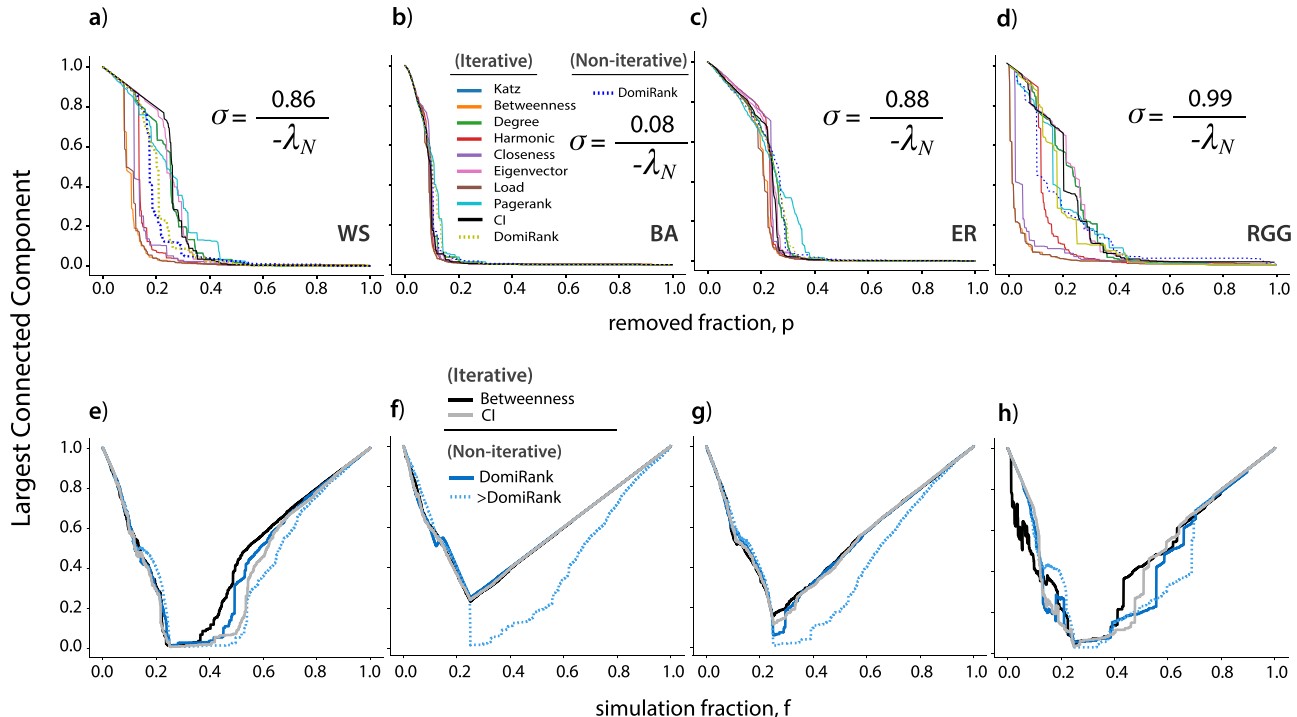

**Fig. 6 | Assessing the effect of iterative centrality-based attacks and recovery mechanisms on network resilience.** Panels **a**–**d** show the evolution of the relative size of the largest connected component of various synthetic networks of size $N = 500$, namely; **a** Watts-Strogatz (WS; $\bar{k} = 4$), **b** Barabási-Albert (BA; $\bar{k} = 4$), **c** Erdős-Rényi (ER; $\bar{k} = 4$), and **d** random geometric graph (RGG; $\bar{k} = 7$), undergoing sequential node removal based on iteratively computed centralities and on pre-computed DomiRank. Panels **e**–**h** show the evolution of the relative size of the largest connected component for the same networks undergoing sequential node removal based on pre-computed DomiRank (optimal and high $\sigma$), iterative betweenness, and Collective Influence (CI), where a stochastic first-in-first-out node recovery (stack recovery implementation) process, with a probability of recovery $p = 0.25$ per time step, is implemented.

information regarding the neighboring structure of the nodes. Despite both CI and DomiRank having the ability to combine local and global network properties, they differ fundamentally in their mechanism to do it, and therefore, their nodal importance assessment could differ significantly (See section S-VI in the SM for more details). In fact, we expect DomiRank (particularly for large $\sigma$) to generally inflict more enduring damage than CI since DomiRank focuses more on fragile neighborhoods, whose fragmentation depends on joint sets of nodes, and once removed, the restitution of a fraction of those nodes might not serve to recover a proportionally equivalent fragmented section.

The first indirect piece of evidence supporting the hypothesis that DomiRank inflicts more severe and enduring damage than other centrality-based attacks is that DomiRank-based attacks remove links more efficiently than other attack strategies (see section S-VII in the SM). To test the hypothesis more directly, we implemented two simple recovery mechanisms to evaluate from which of the attacks the network was less prompt to recover. Both recovery mechanisms assign a probability $p$ to a given removed node to recover every time step, wherein the first strategy selects the nodes in the same order that they were removed (results shown in Fig. 6e–h), while for the second strategy, nodes are selected at random from the pool of removed nodes (see results in section S-VIII in the SM). Our results show that for all the networks, except for the random geometric graph (probably due to network modularity), when a recovery mechanism is put in place, the attack based on a single computation of DomiRank centrality has a comparative dismantling ability than the attack based on iterative betweenness, as shown by the deterioration trend of the *LCC* in Fig. 6e–h. Moreover, for all the analyzed topologies, the DomiRank-based attack causes longer-lasting effects, as the recovery mechanism requires a larger fraction of reinstated nodes to obtain an equivalent recovery in terms of *LCC*. The superior ability of the DomiRank strategy to inflict more severe damage is grounded in its aim to dismantle the

inherent network structure via the dominance mechanism. To further demonstrate this point, Fig. 6e–h also displays a high-$\sigma$ DomiRank-based attack (boosted dominance), where the pace at which the networks recovered was increasingly impeded. Thus, the DomiRank centrality provides a trade-off between the celerity and the severity of the attack through modulation of $\sigma$, highlighting its applicability to design vaccination schemes and other mitigation strategies.

Sequential node failure caused by random or targeted attacks can compromise not only the structure but also the dynamics taking place on the network, i.e., the functional robustness of the network. Here, we benchmark the ability of DomiRank-based attacks to disrupt a rumor-spreading dynamic[55] on different network topologies. We implement an epidemic-like model for spreading rumors, where each node represents an individual who can be in three potential states with respect to the rumor: ignorant, active spreader, and stifler (have heard the rumor but is no longer spreading it)[56]. More specifically, the rumor-spreading dynamic takes four arguments: (i) the network $\mathcal{N}$, (ii) the origin of the rumor (node), (iii) the probability of persuading someone of the rumor ($\rho_r$), and (iv) the probability of becoming a stifler ($\rho_s$). We implement this model on the subsequent networks originating from sequences of node removal according to different centrality-based targeted attacks, choosing the fraction of the population that believes the rumor at the end of the process as the proxy for functional robustness. Specifically, we contrast the results obtained from using a DomiRank-based attack with three other relevant centralities: Degree, PageRank, and CI. The selection of these three metrics is based on different reasons. Degree is a simple, widely-used centrality, which at the same time corresponds to the limit of no competition for DomiRank ($\sigma = 0$). PageRank is arguably one of the most effective metrics to identify critical nodes, as shown in the results in Fig. 5. We also include CI as an example of a sequentially recomputed centrality (after each nodal removal) that is particularly relevant to information-spreading dynamics.

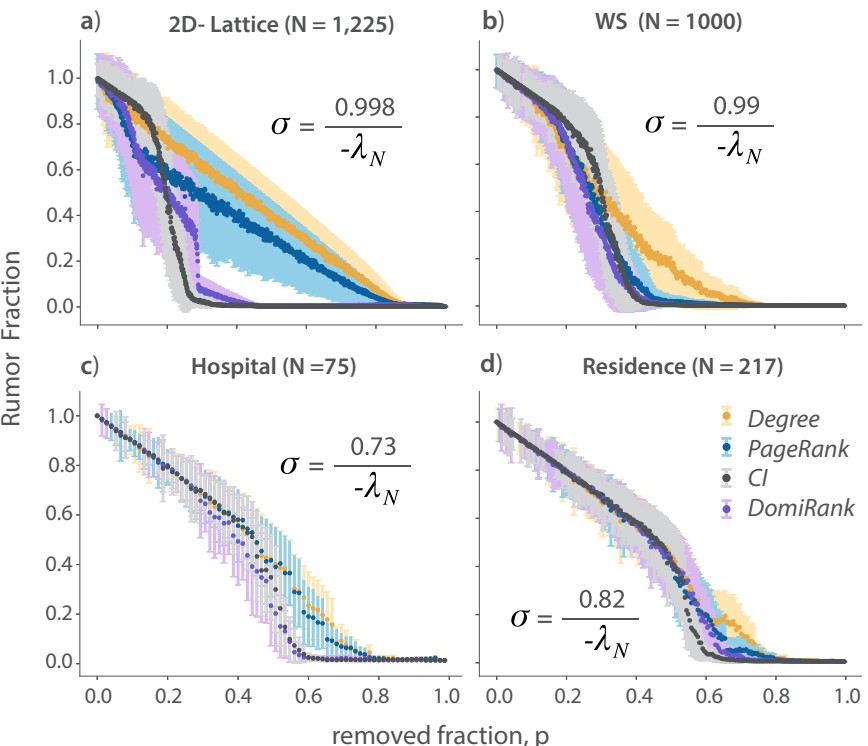

**Fig. 7 | Functional robustness of synthetic and real-world networks under centrality-based attacks.** Average rumor spread fraction (error-bars representing the standard deviation) of 1000 rumor spreading simulations as a function of the subsequent network stage resulting of sequential node removal according to degree, PageRank, DomiRank, and Collective Influence strategies, for two synthetic networks: **a** 2D regular lattice ($\bar{k} = 4$) and **b** Watts-Strogatz (WS; $\bar{k} = 6$), and two real networks: **c** a contact-tracing social network (Hospital; $\bar{k} = 30$) and **d** a survey based social network (Residence; $\bar{k} = 16$).

Figure 7 showcases the superior ability of the DomiRank-based attacks to halt a rumor-spreading process in comparison with degree- and PageRank-based attacks for synthetic and real networks. When compared with CI, a general tendency emerges: DomiRank outperforms in terms of disrupting the spreading process for smaller fractions of nodes removed, but as more nodes are removed, CI becomes competitive or even outperforms DomiRank (see Fig. 7). We attribute this to the fact that DomiRank attempts to halt the spreading process by creating obstructions for the spreading process by removing sets of locally dominant nodes in the network, which very effectively impedes the spreading of information in their neighborhoods. In other words, contrasting with the goal of CI to undermine the multiplicative contagion effect by targeting the removal of potential super-spreaders, DomiRank-based attacks aim to stall the spreading process by fragmenting the spreading domain, quasi-isolating neighborhoods in the network.

The ability of DomiRank to highlight the set of nodes to effectively establish firewalls to mitigate the propagation of rumors is conceptually generalizable to other dynamic processes, such as information transport or epidemic spreading, to name a few, prompting the idea that the DomiRank could be used for establishing efficient vaccination schemes.

## Discussion

This work presents a new centrality metric, called DomiRank, which evaluates nodal importance by integrating different aspects of the network's topology according to a single tunable parameter that controls the trade-off between local (nodal) and mesoscale (structural) information considered. Thus, the competition mechanism embedded in the definition of DomiRank offers an alternative perspective on identifying highly important nodes for network functionality and integrity by contextualizing the relevance of nodes in their respective

neighborhoods, taking into account emergent synergies between not directly connected nodes over overlapping neighborhoods (i.e., joint dominance).

One key feature of DomiRank centrality is its low computational cost and fast convergence. On this front, the DomiRank centrality is competitive with the PageRank centrality whilst being parallelizable, which allows for efficient execution on GPU infrastructure, making it applicable on massive sparse networks.

We show the superior ability of DomiRank to generate effective targeted attacks to dismantle the network structure and disrupt its functionality, offering an outstanding trade-off between the celerity and the severity of the attack and, therefore, significantly reducing network resilience. DomiRank could be further customized to account for localizing heterogeneity in the topology of massive real-world networks, enhancing the assessment of nodal importance in such systems. Also, we anticipate that hybrid attack strategies, where DomiRank is recomputed at different stages of the attack process, might also increase its performance. Moreover, analyzing the robustness of networks in the light of the recently introduced Idle Network (connectivity of the removed nodes by an attack)[46,57] could be particularly illuminating as the DomiRank's parameter exerts a direct control on the fragmentation of the Idle network.

Finally, we want to highlight the broad applicability of DomiRank centrality to different domains, as via its versatile dominance mechanism, it is anticipated to be instrumental for tasks as diverse as improving SPAM detection, establishing effective vaccination schemes, or assessing vulnerabilities in transportation networks, just to name a few. Thus, DomiRank, by revealing fundamental aspects of network fragility, can spur further research to develop more effective mitigation strategies to improve our overall understanding of complex systems structure and resilience.

## Methods

DomiRank centrality assigns a dominance score to nodes in a network based on a competition dynamic, wherein the level of competition is modulated by the parameter $\sigma$. In the Results section, DomiRank centrality was benchmarked versus other centralities by comparing the efficacy of their corresponding attacks. In those cases, DomiRank was computed with the so-called optimal competition level $\sigma^*$, which is the level of competition that generates the most efficient attack (i.e., minimize the area under the curve of the largest connected component for a network undergoing sequential node removal). In this section, we present the methodology used to compute DomiRank Centrality and explore the parameter space to find the optimal level of competition $\sigma^*$.

First, we present the analytical and numerical methods to compute DomiRank centrality. Recall that DomiRank is the steady-state solution to Eq. (1) (with $\sigma = \frac{\alpha}{\beta}$),

$$\frac{1}{\beta}\frac{d\boldsymbol{\Gamma}(t)}{dt} = \sigma A(\theta \boldsymbol{1}_{N\times 1} - \boldsymbol{\Gamma}(t)) - \boldsymbol{\Gamma}(t).$$

The steady-state solution $\boldsymbol{\Gamma}$ exists if $\boldsymbol{\Gamma}(t)$ converges to $\boldsymbol{\Gamma}$,

$$\lim_{t\to\infty}\frac{d\boldsymbol{\Gamma}(t)}{dt} = 0, \tag{8}$$

which implies

$$\lim_{t\to\infty}[\sigma A(\theta \boldsymbol{1}_{N\times 1} - \boldsymbol{\Gamma}(t)) - \boldsymbol{\Gamma}(t)] = 0. \tag{9}$$

We can solve eq. (9) in the following manner,

$$\sigma\theta A\boldsymbol{1}_{N\times 1} - \lim_{t\to\infty}[(\sigma A + I_{N\times N})\boldsymbol{\Gamma}(t)] = 0, \tag{10}$$

and therefore, the analytical solution to DomiRank takes the form:

$$\lim_{t\to\infty}\boldsymbol{\Gamma}(t) := \boldsymbol{\Gamma} = \sigma\theta(\sigma A + I_{N\times N})^{-1}A\boldsymbol{1}_{N\times 1}. \tag{11}$$

Note that the analytical solution involves a term corresponding to the inverse of a sparse matrix. Computing this inverse could pose challenges in terms of both computational time cost and memory usage, especially for increasingly large networks. Alternatively, DomiRank can also be found as the solution of the linear system of equations $(\sigma A + I_{N\times N})\boldsymbol{\Gamma} = \sigma\theta A\boldsymbol{1}_{N\times 1}$, allowing us to obtain the exact calculation of DomiRank for significantly larger networks than when using the inverse formulation. Nevertheless, for massive networks, the computational time cost for the analytical solution may still remain infeasible. For this reason, we introduced the numerical (recursive) solution to DomiRank as displayed in Eq. (6) (with $\sigma = \frac{\alpha}{\beta}$, $\beta = 1$ and without loss of generality),

$$\boldsymbol{\Gamma}(t + dt) = \boldsymbol{\Gamma}(t) + (\sigma A(\theta \boldsymbol{1}_{N\times 1} - \boldsymbol{\Gamma}(t)) - \boldsymbol{\Gamma}(t))dt, \tag{12}$$

allowing the approximation of DomiRank with a computational cost of $\tau(m + 5N)$, where $\tau$ is the total number of time steps required for convergence (set such that $\frac{1}{N}||\boldsymbol{\Gamma}(t) - \boldsymbol{\Gamma}(t + dt)||_1 < dt \cdot \epsilon$), $m$ is the number of links, $N$ is the number of nodes, and computational complexity scaling with $\mathcal{O}(m + N)$.

Now that the different methodologies to compute DomiRank are stated, we focus on estimating $\sigma^*$. We systematically explore the levels of competition $\sigma$, by linearly discretizing the interval of convergence $(0, \frac{-1}{\lambda_N})$ into $n$ (typically $n = 100$) values. For each of these values, we compute DomiRank and generate an attack strategy, wherein nodes are sequentially removed in decreasing order of DomiRank. The effect of each attack strategy is summarized in a curve representing the evolution of the largest connected component throughout the attack.

We use the area under this curve as a quantifier to assess the efficacy of the attack, and thus, $\sigma^*$ is selected as the value of $\sigma$ that produces the attack sequence that minimizes the aforementioned area. Note that both the computation of DomiRank for the different values of $\sigma$, and the size of largest connected component at the different stages of the corresponding DomiRank-based attacks are independent and can thus be computed in parallel. However, it is also important to notice that the computation of the subsequent values of the largest connected component size of a network undergoing an attack is computationally expensive. Specifically, the computation of the largest connect component of a network with $N$ nodes and $m$ links scales with $\mathcal{O}(m + N)$. This computation is repeated after each removal, i.e., $N$ times, yielding a total computational cost $< \mathcal{O}(N(m + N))$. For large networks, we can reduce the computational cost of finding $\sigma^*$ by sampling, i.e., computing the largest connected component every $\frac{1}{\gamma} = 1\%$ of nodes removed. This reduces the computational cost of the evaluation of the attack, effectively causing this computation to scale linearly with the number of edges $< \mathcal{O}(\gamma(m + N))$ in the network. The Results section shows how this methodology can be applied to massive networks of size $N > 20,000,000$.

The implementation in Python can be found by referring to the Code Availability section, or on GitHub (https://github.com/mengsig/DomiRank).

### Reporting summary

Further information on research design is available in the Nature Portfolio Reporting Summary linked to this article.

## Data availability

The data supporting the findings of this study are available within the paper and references.

## Code availability

The code developed to compute DomiRank centrality is available at https://doi.org/10.5281/zenodo.8369910.

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

## Acknowledgements

Y.M. was partially supported by the Government of Aragón, Spain and "ERDF A way of making Europe" through grant E36 23R (FENOL), and by Ministerio de Ciencia e Innovación, Agencia Española de Investigación (MCIN/AEI/10.13039/501100011033) Grant No. PID2020 115800GB I00. A.T. thanks the Spanish Ministry of Universities and the European Union Next Generation EU/PRTR for their support through the Maria Zambrano program. A.T. and E.F-G. were partially supported by NSF (EAR1811909) and the United Kingdom Research & Innovation Living Deltas Hub

NES008926. E.F.-G. also acknowledges partial support by NASA (Grants 80NSSC22K0597 and 80NSSC23K1304) and NSF (Grant IIS2324008). The funders had no role in the study design, data collection, analysis, the decision to publish, or the preparation of the manuscript.

## Author contributions

M.E. and A.T. developed the methods. M.E. implemented the code and conducted the experiments. M.E. and A.T. analyzed the data. M.E. and A.T. wrote the initial manuscript. M.E., A.T., Y.M., E.F.-G. and C.K. discussed the results, revised the manuscript, and approved the final version of the manuscript.

## Competing interests

The authors declare no competing interests.
