## [Peer Review File · Nature Communications]

REVIEWER COMMENTS

Reviewer #1 (Remarks to the Author):

Review of “DomiRank Centrality: revealing structural fragility of complex networks via node dominance.”

This is a very interesting paper that develops a novel centrality measure based on the dominance of a node over its neighboring nodes via the distribution of degree and its constraints on formalized competition dynamics. The formalisms are clear and clever, and the simulated comparisons are exhaustive and convincing. I only have three comments, one major and two minor, which I think – if addressed – will make this paper ready for publication.

Major

1. Need for comparison against optimal percolation centrality (the collective influence algorithm). Despite this paper's focus on targeted attacks aimed at dismantling graphs, it is striking that the authors do not compare their method against the collective influence (CI) algorithm (optimal percolation centrality) defined by Morone & Makse (2015, ref. below). CI is designed to optimally identify nodes that would result in the greatest fragmentation of the largest connected component if they were to be removed (i.e., via a simulated attack). Interestingly, their approach appears to differ from DomiRank in that CI often identifies weakly connected nodes (in terms of degree) as particularly influential in terms of constraining percolation processes. In other words, CI might identify conditions under which the “dominated” nodes in terms of the degree distribution may serve as optimal influencers for maintaining the integrity of the graph. It may be that DomiRank helps to explain the surprising results reported by Morone & Makse, namely that “a large number of previously neglected weakly connected nodes emerges among the optimal influencers. These are topologically tagged as low-degree nodes surrounded by hierarchical coronas of hubs, and are uncovered only through the optimal collective interplay of all the influencers in the network.” Yet, it also remains possible that these algorithms identify different topological signatures of fragility, one performing better than the other in certain topologies, etc. To maximize the impact and relevance of DomiRank, an extensive comparison against optimal percolation centrality is highly recommended. See Pie et. al. (2018) for discussion of the CI algorithm; it too leverages the degree distribution in a not unrelated way, i.e., by measuring the product of the reduced degree centrality of a node and the total reduced degree centrality of all nodes at a given distance d from the focal node.

References

-Morone, Flaviano, and Hernán A. Makse. "Influence Maximization in Complex Networks through Optimal Percolation." *Nature* 524, no. 7563 (August 2015): 65–68. <https://doi.org/10.1038/nature14604>.

-Pei, Sen, Flaviano Morone, and Hernán A. Makse. "Theories for Influencer Identification in Complex Networks." In *Complex Spreading Phenomena in Social Systems: Influence and Contagion in Real-World Social Networks*, edited by Sune Lehmann and Yong-Yeol Ahn, 125–48. *Computational Social Sciences*. Cham: Springer International Publishing, 2018. https://doi.org/10.1007/978-3-319-77332-2_8.

Minor

2. Implications for rich club networks. The authors propose that DomiRank is an effective measure of (ln. 137 – 140) "collusion (joint dominance) emerging from the synergetic action of several individuals in suppressing the fitness of a common neighbor while incrementing their respective fitness." It would be interesting if the authors could explore (either in discussion but ideally also in simulations) the implications of this result for understanding power dynamics in rich club topologies, with are more commonly identified as providing conditions ripe for collusion and corruption (particularly within the rich club) (see Alstott et al. 2014, ref. below, for example of the continuum of rich club topologies).

References

-Alstott, Jeff, Pietro Panzarasa, Mikail Rubinov, Edward T. Bullmore, and Petra E. Vértes. "A Unifying Framework for Measuring Weighted Rich Clubs." *Scientific Reports* 4, no. 1 (December 1, 2014): 7258. <https://doi.org/10.1038/srep07258>.

3. Identifying conditions under which the periphery can gain collective dominance. Building on point 2. above, I think DomiRank has the potential to reveal something really important theoretically speaking about the distribution of influence and power in networks. Specifically, it would be interesting to explore the conditions under which a sufficiently connected periphery may be able to gain dominance over the hub-positions in a graph. This related to the question about rich club structures above. One way to explore this would be to examine the distribution of Domirank between the core and periphery while studying a range of scale-free networks while tuning the connectedness of the periphery (see holme & kim 2002 for methodology). DomiRank should be designed to have the capacity to observe how increasing the connectedness in the periphery can shift dominance from the core to the periphery. Yet, it's also possible that DomiRank's ability to do this will be constrained by the definition of each node's fitness score as dependent on its number of neighbors (ln 126-127); this may limit the ability for the measure to characterize large-scale dominance patterns among peripheral nodes that maintain low degree but nevertheless may be sufficiently connected to achieve collective influence. This is relevant as well to recent advances in modeling influence in the context of complex contagions (see, for example, Guilbeault & Centola's 2021 measure of complex centrality) which shows that hub nodes are often not the most influential for spreading contagions that require substantial local peer reinforcement, instead

more periphery nodes can gain more influence for these contagions due to the chains of bridges that characterize their community structure.

References

-Holme, Petter, and Beom Jun Kim. "Growing Scale-Free Networks with Tunable Clustering." *Physical Review E* 65, no. 2 (2002): 026107.

-Guilbeault, Douglas, and Damon Centola. "Topological Measures for Identifying and Predicting the Spread of Complex Contagions." *Nature Communications* 12, no. 1 (July 20, 2021): 4430.
<https://doi.org/10.1038/s41467-021-24704-6>.

Reviewer #2 (Remarks to the Author):

DomiRank Centrality: revealing structural fragility of 2 complex networks via node dominance

This paper introduces a new centrality metric - DomiRank - to evaluate the importance of nodes in a network. The idea behind this metric is to identify nodes who are locally dominant. Hence, for example, a node with a high degree that is surrounded by many small nodes is central, whereas a similar degree node but situated in a hub-environment is not as central. The authors examine this new ranking method and show that:

1. It generally outperforms existing centrality measures as a network attack strategy.
2. It achieves similar results not just against structural robustness, but also against the network's functionality, as examined through rumor spreading dynamics.
3. The measure is computationally efficient, allowing to analyze large networks within reasonable timescales.

To be honest, when I first read the abstract my feeling was that this is just another robustness paper - an important topic, but also one that has, in my opinion, reached saturation. This impression has not changed - this is, indeed, a robustness paper, however, it incorporates an interesting approach that is, to my knowledge, outside the beaten path. This is expressed in Eq. (1), which extracts the nodes' centrality by incorporating a dynamic process of competition between each node and its immediate neighborhood. Such approach to find structural centralities is interesting and through provoking, and

also offers several paths for generalization. Therefore, despite the chewed up topic of how to break a network apart, this specific application may actually find a suitable place in Nature Communications.

Before diving into the details - let me summarize in broad strokes my impression of this contribution.

The strong points of the paper:

- (i) Its "dynamic" approach by which to rank the nodes.
- (ii) It is well-written and clear.
- (iii) The authors successfully demonstrate the scalability of their method by analyzing extremely large networks.

The main weak points:

- (i) Some of the observed results are underwhelming. This is not something that can be corrected, I believe, however should be reflected in a more reserved language at times.
- (ii) The current presentation lacks focus on the broader insights on what DomiRank teaches us about network structure.

Below I elaborate on these and other issues that could be improved to strengthen the paper:

Equation (1). This is the central contribution offered in this paper - the dynamic equation that determined the DomiRank centrality Γ of all nodes. I highly recommend to explain this equation and the motivation behind it. While the vector notation is compact and elegant, it also obscures some of the "intuition" behind the proposed dynamics. I suggest to follow the text with a term-by-term breakdown of the different equation components. For example, the first term $A \cdot \theta_{1 \times N}$ is simply the degree vector. So for the i -th node it reads $d\Gamma_i/dt = \theta_{k_i}$. In the latter form the meaning becomes easy to grasp, and hence I think it's best to have this explicitly in the text. This also goes on to the second term, which, in non-vector form reads $-\sum A_{ij} \Gamma_j(t)$, once again makes the competition component between i and its neighbor j intuitively apparent.

On the above note, I think the best way to go is to simply add an illustrative figure that depicts the different forces pulling on each node in this formulation. A picture, may, indeed, be worth 1000 words.

Another illustration that can help gain further intuition is to show for a mid-sized network (say $N \sim 20$), the different snapshots along the path to its disintegration. Let the readers "see" how DomiRank always

picks the optimal next contender for removal. This illustration can be tailored to show us, in practice why, e.g., the best node removal is not necessarily the next hub, or the next bridge (betweenness), but rather some other, less expected, node that is, perhaps low degree, but still locally dominant. The current Fig. 1 aims at something along these lines, but does not show all the way through "why" the DomiRank central nodes are also the ones for optimal network attack.

This brings me to weakness (ii) mentioned above. It seems to me that there is deeper insight into network structure that causes DomiRank to perform well. Indeed, the reader (me) asks why is it important to be locally dominant? Why is the competition described in Eq. (1) relevant for network robustness? This is not immediately evident. In my understanding it has to do with the fact that hubs linked to other hubs are not so important, as their removal does not truly disconnect a chunk of the network. In contrast, if a hub is linked to many peripheral nodes, it becomes crucial, since, once removed, all its neighbors become loosely connected (or outright disconnected) from the main network component. Such discussion will not only contribute to the readers understanding, but will also naturally lead to insight into the kind of networks for which DomiRank is most relevant: One example is modular networks, which is discussed in the text, so nothing missing there. But perhaps also networks with negative degree correlations? Hierarchical networks? Other examples? Such discussion would truly strengthen the paper. A complementary discussion is on the limits where DomiRank becomes similar (or even inferior) to other methods. Once again, I can guess that a random scale-free network, i.e. lacking any meso-scale features apart from degree heterogeneity, will, in the limit $N \rightarrow \infty$, have DomiRank \approx Degree centrality.

Another insight is that, as opposed to, e.g., degree centrality DomiRank seems to capture a non-local node characteristic. Indeed, the dynamics of Eq. (1) explicitly links a node to its direct neighbors. But the fixed-point of this equation captures also indirect effects, characterizing the nodes "placement" in the network. This is an aperture by which to add an interesting discussion. What plot would one present to demonstrate this, and rank the locality vs. globality of different measures. This should also depend on σ , which determines precisely the importance of local ($\sigma \rightarrow 0$) vs. global ($\sigma \rightarrow 1$) node characterization. Dont just show us more and more plots of network robustness. There can be a deeper discussion.

This global aspect seems to be highly expressed in the 2D lattice (Fig. 3d). There als the nodes are locally identical, with the only difference being the distance from the boundaries (am I interpreting this correctly?). Indeed, the results are that DomiRank performed well on the lattice. My guess is that this is precisely because the method is able to detect these global characteristics. Without them, how would there be any way to rank nodes in the lattice - they are, after all, all locally similar. If this understanding is true, however, it also indicates that this is a finite size effect. As N approaches infinity the DomiRank difference between the nodes should flatten out, as the boundary becomes negligible. Is that correct? If yes, it should be discussed.

The method incorporates a tunable parameter σ , which the authors set differently per each network. This is understandable, but, in my view also a potential shortcoming of the method. How would you set σ a priori? If you really seek to attack a network, or at least rank the optimal attack sequence, you may want to select σ in advance, not after the fact.

Speaking of σ - what were the values set in Figure 4?

Results. Finally, in some of the figures the advantage of DomiRank seems marginal (e.g., Fig. 3c, Fig. 4c,k...). This may be rectified to some extent if the authors add my suggested analysis of network topologies where DomiRank is most relevant. This will allow to showcase the "interesting" networks, where results will be significant. It will also help explain why, for example, they are not so impressive against the BA networks (which lack meso-scale structure, but have strong local heterogeneity). In any case, the authors may consider tuning down their language on how DomiRank outperforms all other methods, and acknowledge the cases where this "outperformance" is minor.

The paper's outline follows: 1. Presenting DomiRank, 2. Discussing its computational feasibility, 3. Showing results and discussing network insights. My feeling is that the potential readers will find item 3 much more important and interesting, and consider item 2 as a sideline discussion. If the authors agree with me on this, I recommend to reorganize the outline, first focusing on 3, then discussing item 2. Of course, it's up to the authors to decide on that.

As I indicated in the opening - I find this paper interesting, and think it can potentially lead to interesting generalizations down the road. I am looking forward to reconsidering a revised version.

Small corrections:

Barabasi should appear with an accent on the second a (Barabá\{s}si). See, e.g., lines 247 and 294.

Line 410 "more directly", I suggest to replace with "directly". The qualifier "more" seems to be inappropriate here.

DomiRank is a ranking, hence any rescaling or gauging that preserves the rank order is permitted. Therefore, I suggest to decide whether Γ is set to the range -1 to 1, or 0 to 1. There seems to be an inconsistency between Figs. 1 and 3.

Reviewer #3 (Remarks to the Author):

In this manuscript the authors introduce a new centrality measure, the DomiRank centrality, that exploits a competition mechanism among nodes, regulated by a free parameter σ . By opportunely tuning the free parameter, DomiRank centrality is able to modulate the effect of local vs global properties on the node relevance. The effectiveness in the dismantling of the network obtained with DomiRank-based attack is proven on a wide set of synthetic and real-world networks and it is shown that DomiRank-based attacks outperform other centralities-based attacks both in terms of the largest connected component and in terms of the fraction of links removed. The authors also show that DomiRank estimation allows for parallel computation and that the computational cost of recursively calculating DomiRank is relatively low.

The presented measure is of interest to the field and, overall, the study is exhaustive, the number of simulations and comparisons of the proposed quantity with other centrality measures is large enough to properly support DomiRank effectiveness.

Nevertheless, I have some comments on the presented study that I think should be considered before publication:

- As with the DomiRank centrality, other centrality measures depend on some free parameter. For example, the Katz and PageRank centralities contain a free parameter whose value is chosen before the algorithm is applied and that must be set in the interval between 0 and the inverse of the largest (most positive) eigenvalue of A (in Katz centrality) or between 0 and the inverse of the largest eigenvalue of AD^{-1} , where D is the diagonal matrix with elements $D_{ii} = \max\{k_i^{\text{out}}, 1\}$ (for PageRank centrality). Do the authors explore the range of the free parameter for those centralities to find the optimal value to dismantle the network, as well as they explore the range of σ ? Otherwise, the comparison results would be biased and not properly conducted.

- More insights into the analytical formulation of DomiRank centrality should be provided. How is it related to other measures like Katz or PageRank? Are the values of centrality measured at the limit $\sigma \rightarrow \frac{1}{\lambda_N}$ similar to the ones measured with other methods? Which, if any, pitfalls of existing centrality measures is DomiRank able to solve?

- How is DomiRank analytically defined in directed networks? That is, how is it defined in asymmetric adjacency matrices? The authors introduce the case of directed graphs in section III A, a section related to the evaluation of the proposed measure, whereas a formal definition should be provided in section II.

Another specific remark that would improve paper readability is the following:

- In Fig 3, the evaluated optimal values of σ for the different network types are reported. From those quantities, the reader has a hint on how optimal parameter values are related to different meso- and macroscale structural features. To this aim, the selected σ could be included also in the other figures, for example in Fig 4.

REVIEWER COMMENTS

Reviewer #1 (Remarks to the Author):

Review of "DomiRank Centrality: revealing structural fragility of complex networks via node dominance."

This is a very interesting paper that develops a novel centrality measure based on the dominance of a node over its neighboring nodes via the distribution of degree and its constraints on formalized competition dynamics. The formalisms are clear and clever, and the simulated comparisons are exhaustive and convincing. I only have three comments, one major and two minor, which I think – if addressed – will make this paper ready for publication.

We thank the reviewer for their encouraging comments! Below, we provide a detailed response to their inspiring comments, which have helped us improve our manuscript.

Major

1. Need for comparison against optimal percolation centrality (the collective influence algorithm). Despite this paper's focus on targeted attacks aimed at dismantling graphs, it is striking that the authors do not compare their method against the collective influence (CI) algorithm (optimal percolation centrality) defined by Morone & Makse (2015, ref. below). CI is designed to optimally identify nodes that would result in the greatest fragmentation of the largest connected component if they were to be removed (i.e., via a simulated attack). Interestingly, their approach appears to differ from DomiRank in that CI often identifies weakly connected nodes (in terms of degree) as particularly influential in terms of constraining percolation processes. In other words, CI might identify conditions under which the "dominated" nodes in terms of the degree distribution may serve as optimal influencers for maintaining the integrity of the graph. It may be that DomiRank helps to explain the surprising results reported by Morone & Makse, namely that "a large number of previously neglected weakly connected nodes emerges among the optimal influencers. These are topologically tagged as low-degree nodes surrounded by hierarchical coronas of hubs, and are uncovered only through the optimal collective interplay of all the influencers in the network." Yet, it also remains possible that these algorithms identify different topological signatures of fragility, one performing better than the other in certain topologies, etc. To maximize the impact and relevance of DomiRank, an extensive comparison against optimal percolation centrality is highly recommended. See Pie et. al. (2018) for discussion of the CI algorithm; it too leverages the degree distribution in a not unrelated way, i.e., by measuring the product of the reduced degree centrality of a node and the total reduced degree centrality of all nodes at a given distance d from the focal node.

References

-Morone, Flaviano, and Hernán A. Makse. "Influence Maximization in Complex Networks through Optimal Percolation." *Nature* 524, no. 7563 (August 2015): 65–68. <https://doi.org/10.1038/nature14604>.

-Pei, Sen, Flaviano Morone, and Hernán A. Makse. "Theories for Influencer Identification in Complex Networks." In *Complex Spreading Phenomena in Social Systems: Influence and Contagion in Real-World Social Networks*, edited by Sune Lehmann and Yong-Yeol Ahn, 125–48. Computational Social Sciences. Cham: Springer International Publishing, 2018. https://doi.org/10.1007/978-3-319-77332-2_8.

We would like to thank the reviewer for this comment, as we believe the addition of Collective Influence (CI) has significantly strengthened our work. Conceptually CI and DomiRank have some commonalities. For example, both have a parameter (ℓ for CI and σ for DomiRank) that allows them to tune how much global information should be accounted for in the respective centrality. Thus, both CI and DomiRank converge to degree centrality for the lower bound of their parameters ($\ell = 0$ and $\sigma = 0$), as only local information (degree) is considered. As more global information is accounted for in their computation (higher value of their respective parameters), both metrics are able to generate rankings, wherein apparently unimportant nodes could be ranked highly and vice versa, excelling in designing strategies to dismantle networks and halting their dynamics. However, those rankings differ substantially from each other, as they emerge from very different underlying principles. While DomiRank is grounded in a competition dynamic that allows integrating global information, acknowledging and mining the mesoscale features of networks, CI accounts for the multiplicative cascade of paths that each node could develop at different scales. Notably, they also differ in the fact that the efficient attack based on CI fundamentally rests on its iterative nature (it is recomputed after each nodal removal), while the competition mechanism of DomiRank inherently provides scores informed by the values of its neighborhood, permitting to assign widely disparate values for neighboring nodes by a single calculation of DomiRank (non-iterative).

To gain further insight, draw parallels, and highlight the differences between DomiRank and CI, we have explored those centralities in dismantling several toy networks.

- The case of the 2D square lattice network is particularly illuminating. We observe that both CI (Fig R1a) and DomiRank (see Fig R1b) are able to mine the structural symmetries (global information) of the regular network to propose nodal rankings, which lead to dismantling strategies (attacks) significantly more effective than the rest of the non-iterative centralities (See Fig R1d). DomiRank, via its embedded competition mechanism, creates an alternating spatial pattern of high-low ranks in its pre-computed version, which is particularly efficient in designing attack strategies from the very early stages of removal. A different alternating spatial pattern arises from CI, which in this case lies on the iterative nature of this centrality (compare FigR1a for iterative, and FigR1c for non-iterative), *i.e.*, as the most central node in terms of CI is removed, its immediate neighbors drop their CI score. Thus, both centralities are able to exploit the underlying *fragility* associated with the structure of the network via different mechanisms but propose almost perfectly antagonistic ranking assessments highlighting their different approaches to integrate global information.

Figure R1 – Collective Influence and DomiRank on a regular 2D lattice. A comparison between three centralities on a 2D regular lattice ($N=49$), (a) Collective Influence, (b) DomiRank, and (c) non-iterative Collective Influence, along with (d) the evolution of the relative size of the largest connected component whilst undergoing sequential node removal according to various centralities. *Note that this figure has been included in the SM as Fig. S7.*

- Another interesting case to gain insight is a random network (e.g., Erdős-Rényi), where a mix of local and mesoscopic features might emerge. In Figure R2, we show the node centralities (CI – top panels and DomiRank – Bottom panels) for a toy network generated using the Erdős-Rényi model. For this network, there is an agreement in the classification of the two most central nodes, as these nodes are key for information spreading whilst simultaneously dominating their neighborhood. However, the differences between DomiRank and CI emerge in the following node removals, as shown by Fig. R2(b,e). Here, CI attempts to remove the most influential nodes, effectively mitigating the potential cascade originating from this node. On the other hand, DomiRank, rather than removing the most influential nodes, creates obstacles for the spreading process, effectively hindering the potential cascade whilst destroying the periphery of the removed nodes -

i.e., continuously dismantling the structure of the network. Fig. R2(d-f) reveals that DomiRank attempts to fragment the network into small components systematically by placing importance on nodes having a weakly connected periphery. Finally, Fig. R2g shows that despite the substantially different mechanisms in their evaluation of important nodes, the ability to dismantle the network remains similar, outperforming one another at various stages of the attack until they both reach optimal percolation of the cluster of size $\ell + 1$, despite having removed a substantial amount of different nodes.

Figure R2 - Collective Influence and DomiRank on a toy random network. Comparison between (a,b,c) Collective Influence and (d,e,f) DomiRank and how they dismantle a toy Erdős-Rényi network $N = 32$ when nodes are sequentially removed according to descending values of their centralities. (g) The evolutions of the largest connected component of the network undergoing those attacks, along with various other centralities for reference, are also shown. Note that the color scale used to represent the relative values of the centralities in panels (a-f) is re-normalized for each panel for visualization purposes. *Note that this figure has been included in the SM as Fig. S9.*

- We have also included a toy model of a hierarchical structure like the ones commented by the reviewer (Figure R3), and indeed, we confirm the reviewer's hypothesis that nodes surrounded by hierarchical coronas would be highlighted as very central nodes by CI, while DomiRank would tend to tag them as dominated nodes (low score). More generally, the nodal importance assessment of DomiRank for this type of node, as highlighted in the two previous toy network examples, would depend on the networks' structural properties, particularly those coronas, as highlighted by the relative emergent pattern of dominance. In the case of the central node shown in Figure R3, CI and DomiRank diverge in their assessment of their importance, as CI assigns it the highest score while DomiRank the lowest. Thus, DomiRank-based attacks would sequentially remove the hierarchical corona of hubs surrounding the low-degree node, which satisfies two objectives: removing network hubs and continuously decreasing the importance of the central node by isolating it - paraphrasing it using CI terminology, DomiRank attempts to remove influential nodes (not the top influencer), while reducing the influence of the top influencer. On the other hand, CI-based attacks would first remove the low-degree node, which impedes the

connectivity between hubs, and then sometime later, the CI-based attack would remove the hubs as they are important due to their high-degree, having particular relevance for spreading processes on networks.

Fig R3 - Collective Influence and DomiRank on a network consisting of a corona of hubs. The relative values of (a) Collective Influence and (b) DomiRank nodal importance are displayed by the coloring of the nodes. It is worth noting that the relative ranking of nodes according to DomiRank is quite independent of the value of σ for this configuration. *Note that this figure has been included in the SM as Fig. S8.*

Obviously, we were also excited to perform a comparative analysis of DomiRank vs. CI for the other networks included in the paper! Now, those results are part of the amended manuscript. In particular, we have included CI in the comparative analysis of attacks based on iterative centralities performed for larger scale ($N = 500$) synthetic networks for diverse topologies (see Fig 6 in the MS – reproduced here as Fig R4 for completeness). The results align with the insight provided with toy networks, observing a general tendency that for a small number of node removals, DomiRank-based attacks tend to outperform CI-based attacks. However, for some network topologies, the CI-based attacks become more effective after a certain fraction of node removals. We have also compared the performance of CI in the case where a nodal recovery mechanism is in place. We show here the results in Fig R4-bottom panels (also included in Fig 6 in the MS), where DomiRank-based attacks are still causing more enduring damage in the network structure as recovery processes are comparatively slower to heal the system. That being said, CI also causes more enduring damage than iterative betweenness, providing evidence supporting that CI, whilst removing the most important influencers, also damages the network structure due to its iterative recomputation that creates relatively disparate scores between adjacent nodes.

Fig R4 - Assessing the effect of iterative centrality-based attacks and recovery mechanisms on network resilience. Panels a-d show the evolution of the relative size of the largest connected component of various synthetic networks of size $N=500$, namely, (a) Watts-Strogatz (WS; $\bar{k} = 4$), (b) Barabási-Albert (BA; $\bar{k} = 6$), (c) Erdős-Rényi (ER; $\bar{k} = 5$), and (d) random geometric graph (RGG; $\bar{k} = 7$), undergoing sequential node removal based on iteratively computed centralities and on pre-computed DomiRank. Panels e-h show the evolution of the relative size of the largest connected component for the same networks undergoing sequential node removal based on pre-computed DomiRank (optimal and high σ), iterative betweenness, and Collective Influence (CI), where a stochastic first-in-first-out node recovery (stack recovery implementation) process with a probability of recovery $p = 0.25$ each time step is implemented. *Note that this figure has been included in the MS as Fig. 6.*

We have also performed a comparative analysis of DomiRank and CI in terms of functional robustness. From this perspective, both metrics excel in halting spreading processes on the networks but with different mechanisms. DomiRank-based attacks remove dominant nodes in a network, very effectively impeding the spreading of information in the neighborhood of the removed node; in other words, DomiRank attempts to place highly effective local obstacles rather than attempting to stop the dynamics of the whole system by finding the potential most influential spreader. On the other hand, CI fundamentally seeks the most effective spreader, and thus, CI-based attacks aim to undermine the multiplicative contagion effect of the spreading process. These differences lead to different efficiencies of the attacks depending on their stage, generally obtaining more effective DomiRank-based attacks in halting the spreading at earlier stages of the attack strategies, while CI-based attacks require a smaller fraction of nodes removed to inhibit the process completely (See Fig R5).

Fig R5 - Functional robustness of synthetic and real-world networks under centrality-based attacks. Average rumor spread fraction (error-bars representing the standard deviation) of 1000 rumor spreading simulations as a function of the subsequent network stage resulting in sequential node removal according to degree, PageRank, DomiRank, and Collective Influence strategies for two synthetic networks: (a) 2D regular lattice ($\bar{k} = 4$) and (b) Watts-Strogatz (WS; $\bar{k} = 6$), and two real networks: (c) a contact-tracing social network and (d) a survey-based social network. *Note that this figure has been included in the MS as Fig. 7.*

Once more, we thank the reviewer for this suggestion, and we want to note that we have made substantial changes in the manuscript to reflect the insight gained from the comparison with CI. Particularly, we want to highlight the following changes in the paper and the SM:

- i. We have added a section (SM-VI) in the SM that compares CI with DomiRank.
- ii. We have included CI in the analysis presented in Fig. 6 in the manuscript.
- iii. We have included CI in the functional robustness analysis in the manuscript (Fig. 7) due to its particular importance in spreading processes.
- iv. We have changed numerous parts of the text to give further insight into DomiRank, both in light of the understanding gained from the conceptual comparison of CI and making comparisons to CI regarding the modified figures.

Minor

2. Implications for rich club networks. The authors propose that DomiRank is an effective measure of (ln. 137 – 140) "collusion (joint dominance) emerging from the synergetic action of several individuals in suppressing the fitness of a common neighbor while incrementing their respective fitness." It would be interesting if the authors could explore (either in discussion but ideally also in simulations) the implications of this result for understanding power dynamics in rich club topologies, with are more commonly identified as providing conditions ripe for collusion and corruption (particularly within the rich club) (see Alstott et al. 2014, ref. below, for example of the continuum of rich club topologies).

References

-Alstott, Jeff, Pietro Panzarasa, Mikail Rubinov, Edward T. Bullmore, and Petra E. Vértés. "A Unifying Framework for Measuring Weighted Rich Clubs." *Scientific Reports* 4, no. 1 (December 1, 2014): 7258. <https://doi.org/10.1038/srep07258>.

3. Identifying conditions under which the periphery can gain collective dominance. Building on point 2. above, I think DomiRank has the potential to reveal something really important theoretically speaking about the distribution of influence and power in networks. Specifically, it would be interesting to explore the conditions under which a sufficiently connected periphery may be able to gain dominance over the hub-positions in a graph. This related to the question about rich club structures above. One way to explore this would be to examine the distribution of Domirank between the core and periphery while studying a range of scale-free networks while tuning the connectedness of the periphery (see holme & kim 2002 for methodology). DomiRank should be designed to have the capacity to observe how increasing the connectedness in the periphery can shift dominance from the core to the periphery. Yet, it's also possible that DomiRank's ability to do this will be constrained by the definition of each node's fitness score as dependent on its number of neighbors (ln 126-127); this may limit the ability for the measure to characterize large-scale dominance patterns among peripheral nodes that maintain low degree but nevertheless may be sufficiently connected to achieve collective influence. This is relevant as well to recent advances in modeling influence in the context of complex contagions (see, for example, Guilbeault & Centola's 2021 measure of complex centrality) which shows that hub nodes are often not the most influential for spreading contagions that require substantial local peer reinforcement, instead more periphery nodes can gain more influence for these contagions due to the chains of bridges that characterize their community structure.

References

-Holme, Petter, and Beom Jun Kim. "Growing Scale-Free Networks with Tunable Clustering." *Physical Review E* 65, no. 2 (2002): 026107.

-Guilbeault, Douglas, and Damon Centola. "Topological Measures for Identifying and Predicting the Spread of Complex Contagions." *Nature Communications* 12, no. 1 (July 20, 2021): 4430. <https://doi.org/10.1038/s41467-021-24704-6>.

We would like to thank the reviewer for yet another very insightful comment. Note that since the points 2 and 3 made by the reviewer relate to each other, we took the liberty of answering both together.

Indeed, describing Rich-Club power dynamics seems intuitive and natural in terms of DomiRank. Thus, we have included some discussion about the implication of the dominance dynamics in Rich-Clubs that we believe add value and insight to our work. In this response and the SM, we present some preliminary simple experiments to highlight the specific insight gained from the study of Rich-Clubs networks through the lens of DomiRank. Actually, this preliminary analysis has spurred great excitement in continuing our research in this domain by comprehensively and systematically exploring Rich clubs in terms of DomiRank, as we believe that it will reveal important properties of such networks.

In the first part of our analysis, we have developed a small toy model of Rich-Club networks (see Fig R6a), which consists of an interconnected Rich Club of hub nodes. Each hub node (leader) in the Rich Club is a star network with a varying number of nodes attached to it (periphery). Particularly, we have included six star networks with varying degrees, connecting them in a simple ring. This example shows that despite the simple structure, where apparently, degree emerges as the only defining property, DomiRank can reveal important properties relevant to power dynamics. In particular, the example included in the SM guides the reader through some targeted modifications of the connectivity of the hubs and their peripheries and serves to illustrate the underlying mechanisms controlling and steering nodal dominance. We reproduce the text included in the SM here for completeness of this response, advancing that the rationale and intuition shown in the reviewer's comments were indeed correct.

Fig R6 - Evolution of dominance in a Rich-Club network by altering connectivity. A case study of the evolution of dominance (color-encoded) in a (a) simple Rich-Club network, when new competitions (links) are established in the Rich-Club and their respective peripheries. The evolutionary cases have the following iterative links introduced: (b) link (1,3) and link (1,5); (c) link (2,5); (d) link (3,6); (e) link between the periphery of node 3 and the periphery of node 5; and (f,g,h) internal competition (addition of links) in the periphery of node 5. All DomiRank centralities are computed for $\sigma = \frac{0.99}{-\lambda_N}$, in order to simulate a highly competitive environment where changes in power dynamics are more likely to occur. *Note that this figure has been included in the SM as Fig. S1.*

“Fig. S1a [here Fig R6a] displays the chosen initial configuration of the Rich-Club, where node coloring represents nodal dominance given by DomiRank centrality ($\sigma = \frac{0.99}{-\lambda_N}$), and nodes are arbitrarily labeled with numbers to ease the discussion that follows. Here, node 1 is dominating (highest DomiRank score) as it has the largest periphery, and additionally, it is jointly dominating nodes 2 and 4, by colluding with nodes 3 and 5, respectively. Thus, nodes 3 and 5 exhibit dominant behavior, although below the levels obtained by node 1. Actually, node 1 can reinforce its dominant role by establishing direct competition (links) with nodes 3 and 5 (Fig. S1b [here Fig R6b]). In such a scenario, node 1 is able to subdue nodes 3 and 6 with the help of node 4, which contributes via joint dominance. A more drastic power shift such that nodes 3 and 5 jointly dominate the rich club can also be induced, by establishing competition between nodes 5 and 2, and 3 and 4 (Fig. S1c,d [here Fig R6c,d] respectively).

Furthermore, changes in the relative dominance of the nodes in the rich club can be promoted by modifying the connectivity of their peripheries. Thus, the dominance of node 3 over node 5 can be induced by creating competition in the periphery of node 5, which would internally challenge the dominance of node 5 (Fig. S1e-g [here Fig R6e-g]). In other words, an increase in the connectivity (competition) of the periphery reduces the relative dominance of the center node connected to that periphery (node 5), making its periphery less dominated and more “fit”. The power shift observed from introducing competition in the periphery of a hub is not as drastic as that of creating direct competition for members within the Rich Club; however, it is sufficient to steer the power balance within the Rich Club.

These simple examples show how DomiRank is able to naturally describe collusion scenarios where nodes can dominate neighborhoods and gain power by establishing common enemies. Therefore, we show that we can steer power dynamics by artificially forcing competition and joint dominance between Rich-Club nodes, or we can ruin the dominance of a Rich-Club node, by creating anarchy in its periphery.”

The new insight gained from Rich-Club networks is reflected now in our work in the following instances:

- I. We have conceptually contextualized DomiRank interpretation in terms of Rich-clubs in the main manuscript to gain insight into the meaning of dominance and highlight its potential applications (lines 162-174)
- II. We have included a new section in the SM (SM-I), that aims to show with an illustrative network example the controlling factors underlying power shifts in Rich-Club networks through the lens of nodal dominance (DomiRank).

Reviewer #2 (Remarks to the Author):

DomiRank Centrality: revealing structural fragility of 2 complex networks via node dominance

This paper introduces a new centrality metric - DomiRank - to evaluate the importance of nodes in a network. The idea behind this metric is to identify nodes who are locally dominant. Hence, for example, a node with a high degree that is surrounded by many small nodes is central, whereas a similar degree node but situated in a hub-environment is not as central. The authors examine this new ranking method and show that:

1. It generally outperforms existing centrality measures as a network attack strategy.
2. It achieves similar results not just against structural robustness, but also against the network's functionality, as examined through rumor spreading dynamics.
3. The measure is computationally efficient, allowing to analyze large networks within reasonable timescales.

To be honest, when I first read the abstract my feeling was that this is just another robustness paper - an important topic, but also one that has, in my opinion, reached saturation. This impression has not changed - this is, indeed, a robustness paper, however, it incorporates an interesting approach that is, to my knowledge, outside the beaten path. This is expressed in Eq. (1), which extracts the nodes' centrality by incorporating a dynamic process of competition between each node and its immediate neighborhood. Such approach to find structural centralities is interesting and through provoking, and also offers several paths for generalization. Therefore, despite the chewed up topic of how to break a network apart, this specific application may actually find a suitable place in Nature Communications.

Before diving into the details - let me summarize in broad strokes my impression of this contribution.

The strong points of the paper:

- (i) Its "dynamic" approach by which to rank the nodes.
- (ii) It is well-written and clear.
- (iii) The authors successfully demonstrate the scalability of their method by analyzing extremely large networks.

The main weak points:

- (i) Some of the observed results are underwhelming. This is not something that can be corrected, I believe, however should be reflected in a more reserved language at times.
- (ii) The current presentation lacks focus on the broader insights on what DomiRank teaches us about network structure.

Below I elaborate on these and other issues that could be improved to strengthen the paper:

We would like to thank the reviewer for their compliments, intuitive summary, and important insights.

Equation (1). This is the central contribution offered in this paper - the dynamic equation that determined the DomiRank centrality Γ of all nodes. I highly recommend to explain this equation and the motivation behind it. While the vector notation is compact and elegant, it also obscures some of the "intuition" behind the proposed dynamics. I suggest to follow the text with a term-by-term breakdown of the different equation components. For example, the first term $\sum_{j=1}^N k_{ij}$ is simply the degree vector. So for the i -th node it reads $d\Gamma_i/dt = \sum_{j=1}^N k_{ij} \Gamma_j(t) - \Gamma_i(t)$. In the latter form the meaning becomes easy to grasp, and hence I think its best to have this explicitly in the text. This also goes on to the second term, which, in non-vector form reads $-\sum_{j=1}^N A_{ij} \Gamma_j(t)$, once again makes the competition component between i and its neighbor j intuitively apparent.

We would like to thank the reviewer for their comment!

We have added equations (2) and (3) in the MS, where the vector notation is translated to nodal notation, as suggested by the reviewer. Naturally, we have also included a complementary text providing the easily accessible insight gained from the nodal notation, by decomposing the equation as suggested by the reviewer. We once again thank the reviewer, as we agree that this notation makes the intuition behind the proposed dynamics substantially easier to grasp. See lines 113-131 in the MS.

On the above note, I think the best way to go is to simply add an illustrative figure that depicts the different forces pulling on each node in this formulation. A picture, may, indeed, be worth 1000 words.

Thanks for this suggestion. We didn't think of it, and it really adds a lot to show the underlying dynamics!

Thus, we have modified Figure 1 in the MS to directly display the direction and magnitude of the interactions between nodes emerging from the competition dynamics at a steady state. We reproduce here Figure 1a from MS as Fig R7 for practical reasons and to ease the discussion. Particularly, we show the direction of each pairwise interaction by the arrow directions and its relative intensity of the interaction ('transfer of fitness') by the link thickness. This representation has allowed us to provide much more insight into the emerging dynamics, for example, how for very competitive environments (high $\sigma = \alpha/\beta$ - e.g., Fig R7c), negative scores of DomiRank (Γ) emerge, where an individual with a deficit value of fitness is interpreted as fully 'submissive' to the dominant nodes, giving up its resources to adjacent nodes, rather than fighting for those resources (note the change in the direction of the arrows for the squared node in Fig R7c with respect to Fig R7a and Fig R7b). Those nodes with negative scores are able to maintain a steady state value of fitness due to the relaxation mechanism (governed by β), which serves as a recovery/healing mechanism for them.

This discussion has been reflected in the MS in lines 230-243.

Fig R7 - DomiRank for different levels of competition (σ). DomiRank centrality is displayed on the nodes of a simple network with $N = 15$ nodes for (a) low, (b) medium, and (c) large values of σ . Panel d shows the DomiRank centrality as a function of σ , wherein each solid line represents a specific node (color encoding node degree). In panels (a-c), the direction of the pairwise transfer of fitness between nodes is shown by arrows, with their thickness representing the magnitude of that exchange. Note that for visualization purposes, the arrow thickness in panels (a), (b), and (c) are scaled 25:5:1.

Another illustration that can help gain further intuition is to show for a mid-sized network (say $N \sim 20$), the different snapshots along the path to its disintegration. Let the readers "see" how DomiRank always picks the optimal next contender for removal. This illustration can be tailored to show us, in practice why, e.g., the best node removal is not necessarily the next hub, or the next bridge (betweenness), but rather some other, less expected, node that is, perhaps low degree, but still locally dominant. The current Fig. 1 aims at something along these lines, but does not show all the way through "why" the DomiRank central nodes are also the ones for optimal network attack.

We would once again like to thank the reviewer for this great suggestion. We have modified the former Figure 3 in the MS (now Figure 4) by adding, for each of the topologies, particular stages of the attack sequence, where we can exemplify the relation between the underlying mechanism in the DomiRank definition and its effectiveness in dismantling network structure.

We reproduce here the new Figure 4 from the MS as Figure R8 for discussion purposes. For the regular topology (2D- Lattice), where the optimal value of σ ($\sigma = 0.999 / -\lambda_N$) corresponds to a highly competitive environment, we observe how the distinct alternating spatial pattern for DomiRank scores emerges from the competition dynamics and the regular structure of the network. An attack strategy based on such a spatial pattern is significantly advantageous with respect to other traditional centrality-based centralities, as nodes are left in isolation by removing the existing neighbors and reducing the size of LCC efficiently (See Fig R8a-d). For hub-dominated topologies (e.g., Barabási-Albert model – Fig R8 i-l), there is general consensus

reached by all the metrics on the nodes which are more central (the hubs), yet the dominance dynamics, although with moderate competition ($\sigma = 0.5/-\lambda_N$) is able to fragment more meticulously the different clusters by avoiding over-assessments of node importance arising from being neighbors to other important nodes (conditional importance). This point is well illustrated in panels Fig R8j, particularly if the distribution of DomiRank and PageRank scores are scrutinized in detail. Finally, the case of random topology (Fig R8e-h) showcases even better how DomiRank strikes a balance between local importance and global information, dismissing spuriously (conditional) important nodes. This is particularly apparent when Fig R8(f-h) are examined. We also notice here that DomiRank attempts to shatter the network into many small components, truly fragmenting the network - an observation that is consistent across all topologies and occurs as a consequence of adjacent nodes tending to have disparate scores.

FIG. R8. Comparing DomiRank with other centralities on dismantling toy networks. Evolution of the relative size of the largest connected component whilst undergoing sequential node removal according to their descending scores of various centralities for three toy networks: (a) 2D regular lattice ($N = 49$), (e) Erdős-Rényi (ER; $N = 32$), and (i) Barabási-Albert (BA; $N = 25$). For each topology, panels (b-d), (f-h), and (j-l) show the graphical representation of the respective toy networks at various stages of the attack based on DomiRank, betweenness, closeness, and PageRank centralities. Note that the nodes are colored according to the relative value of the centralities normalized to be in an interval $[0, 1]$ for enhancing comparability and visualization purposes. *Note that this figure has been included in the MS as Fig. 4.*

In line with the reviewer's comments, we have modified the former MS Fig. 3 to the new MS Fig. 4 (identical to Fig. R8), and provided a detailed complementary discussion of the new insight gained from the amended figure - see lines 397-437.

This brings me to weakness (ii) mentioned above. It seems to me that there is deeper insight into network structure that causes DomiRank to perform well. Indeed, the reader (me) asks why is it important to be locally dominant? Why is the competition described in Eq. (1) relevant for network robustness? This is not immediately evident. In my understanding it has to do with the fact that hubs linked to other hubs are not so important, as their removal does not truly disconnect a chunk of the network. In contrast, if a hub is linked to many peripheral nodes, it becomes crucial, since, once removed, all its neighbors become loosely connected (or outright disconnected) from the main network component. Such discussion will not only contribute to the readers understanding, but will also naturally lead to insight into the kind of networks for which DomiRank is most relevant: One example is modular networks, which is discussed in the text, so nothing missing there. But perhaps also networks with negative degree correlations? Hierarchical networks? Other examples? Such discussion would truly strengthen the paper. A complementary discussion is on the limits where DomiRank becomes similar (or even inferior) to other methods.

We thank the reviewer for raising this concern, as it allows us to provide more insight and better describe the importance of dominance in revealing the fragility of networks in terms of structure and dynamics. We first want to acknowledge that the reviewer's rationale and intuition displayed in their comment are absolutely right. DomiRank excels in dismantling network structure and dynamics because it assigns particular high scores to well-connected nodes (e.g., hubs) with sparsely connected peripheries. Thus, removing those nodes tends to produce direct fragmentation of those *fragile* neighborhoods. On the other hand, DomiRank is also able to identify fragile structures via the joint dominance mechanism. This mechanism does not rely as significantly on the local properties of the nodes (degree) but on the nodes' position on the global network structure. This source of dominance emerges mostly in highly competitive environments (high σ) wherein a set of nodes is subject to exert joint dominance if they possess some overlapping neighborhood (periphery) and are preferably not directly connected among themselves. Thus, each of those nodes contributes to partially suppressing the dominance in the common neighborhood, which, if it lacks connections to other nodes in the network, would be entirely dominated by the joint set, creating a positive feedback that reinforces the dominance pattern. From the point view of the structural integrity of the network, the described mechanism indeed serves to identify fragile parts of the network since the removal of the dominant joint set would lead to the fragmentation of the dominated shared neighborhood.

Indeed, we totally agree that this new addition proposed by the reviewer really adds context to the reader and strengthens the paper. We have added a new subsection in Section II from the MS titled “Dominance and network fragility”, where we provide the above arguments in detail (see lines 288-324 in the MS).

We also note that given these two alternative sources of fragility that DomiRank is able to identify, and that the centrality parameter (σ) allows modulating the weight of those two sources of fragility in the final scores, DomiRank is fundamentally versatile across different topologies. It is obvious that if the local properties are the underlying fundamental property of topology, there will be little room to gain for DomiRank versus other metrics such as PageRank, Katz, or even degree. Similarly, in scale-free networks with negative degree correlations, the edge of DomiRank over degree will also be reduced, as all hubs will tend to have sparse peripheries. However, while the levels of negative degree correlations are not extremely high, some interesting mesoscale structures will emerge, which could be mined by DomiRank, either through direct or joint dominance. To further consider similarities and differences between the analyzed centralities, we have also included section IV in the SM, where correlation maps between the different metrics are shown to highlight the variability in their potential relations for different topologies.

At this point, we want to emphasize that we truly sympathize with the reviewer's comments about the large number of similar plots displayed in this work (*i.e.*, evolution of the relative size of the largest connected component as a function of the attack stage), where, in some cases, the gain from DomiRank-based attack is not that prominent. However, we believe that the excitement does not emerge from each of these independent plots, but rather, it is the joint message: for very different sizes and topologies, for synthetic and real networks, DomiRank by a little or a substantial margin, is able to systematically outperform all the other metrics, which from our perspective is indeed exciting and outstanding because none of the other metric-based attacks withstand a comparison across such a mix of networks.

Once again, I can guess that a random scale-free network, *i.e.* lacking any meso-scale features apart from degree heterogeneity, will, in the limit $N \rightarrow \infty$, have DomiRank \approx Degree centrality.

The reviewer is absolutely right. Fig R9 shows that for scale-free networks of increasing size, the optimal value of σ tends to zero. Recall also that at the limit of σ equal to zero, DomiRank converges to degree (see eq 3 in the MS).

FIG. R9. Optimal values of σ for scale-free networks of varying size. Optimal σ values are displayed for scale-free networks with sizes spanning six orders of magnitude [$10^1 - 10^7$], showing that asymptotically $\sigma \rightarrow 0$ as $N \rightarrow \infty$.

Another insight is that, as opposed to, e.g., degree centrality DomiRank seems to capture a non-local node characteristic. Indeed, the dynamics of Eq. (1) explicitly links a node to its direct neighbors. But the fixed-point of this equation captures also indirect effects, characterizing the nodes "placement" in the network. This is an aperture by which to add an interesting discussion. What plot would one present to demonstrate this, and rank the locality vs. globality of different measures. This should also depend on sigma, which determines precisely the importance of local (sigma $\rightarrow 0$) vs. global (sigma $\rightarrow 1$) node characterization. Dont just show us more and more plots of network robustness. There can be a deeper discussion.

This global aspect seems to be highly expressed in the 2D lattice (Fig. 3d). There als the nodes are locally identical, with the only difference being the distance from the boundaries (am I interpreting this correctly?). Indeed, the results are that DomiRank performed well on the lattice. My guess is that this is precisely because the method is able to detect these global characteristics. Without them, how would there be any way to rank nodes in the lattice - they are, after all, all locally similar. If this understanding is true, however, it also indicates that this is a finite size effect. As N approaches infinity the DomiRank difference between the nodes should flatten out, as the boundary becomes negligible. Is that correct? If yes, it should be discussed.

We thank the reviewer for this suggestion! As the reviewer correctly states, the value of σ sets up the trade-off between local and global information. We have incorporated a new figure to MS (Figure 2) to illustrate it and enrich the discussion of this point in the text. We reproduce this figure here as Figure R10 to facilitate the discussion.

When σ is close to its lower bound (see Fig R10a), DomiRank converges to degree, and only local information is considered; therefore, all the nodes, but the ones at the boundary of the lattice,

tend to have the same DomiRank. As the value of σ increases, the values of DomiRank start to diverge from the degree scores as the value corresponding to each node is a function of their neighbors' state (fitness). Thus, Figure R10b shows how the nodes directly connected to nodes in the boundary of the lattice can partially dominate them, increasing their DomiRank score (note that this effect appears for smaller values of σ than the one displayed in this panel, but its visualization is less apparent using a consistent color scheme across panels). As σ keeps increasing (Fig R10c), competition dynamics progresses, and more internal nodes are able to *feel* the boundary as its effect is propagated via domination adapting the DomiRank scores. In other words, the score of an internal node is not just a function of the topological features of their immediate neighborhood, but it acknowledges more distant features. As σ reaches its maximum value (Fig R10d), the DomiRank score of each node is partially affected by the rest of the network via the competition mechanism. This effect is particularly apparent in a regular 2d-lattice where the final pattern obtained is driven by two global system properties, namely the finite boundary and global symmetries (as in a lattice with periodic boundary conditions or an infinity lattice all the nodes are indistinguishable from DomiRank perspective).

Figure R10 - DomiRank for different levels of competition (σ). The DomiRank centrality distribution is displayed on the nodes of a 2D Square lattice with $N = 49$ nodes for different values of σ (annotated in the different panels) to illustrate how different levels of competition set the trade-off between local (nodal) and global (meso- to large- scale structure) for DomiRank. Note that for each panel, the values of DomiRank are normalized to range in interval $[0, 1]$ for enhanced visualization. *Note that this figure has been included in the MS as Fig. 2.*

As mentioned, this new figure has been included in the amended MS as Figure 2, and its discussion is reflected in lines 244-287.

The method incorporates a tunable parameter σ , which the authors set differently per each network. This is understandable, but, in my view also a potential shortcoming of the method. How would you set σ a priori? If you really seek to attack a network, or at least rank the optimal attack sequence, you may want to select σ in advance, not after the fact.

Fair point!

As a preamble to the response to this concern, we want to emphasize how the sensitivity of DomiRank to its unique parameter σ apparent from analyzing different topologies, far from being a weakness, is a key strength of DomiRank, as it allows us to assess the important nodes in networks with topologies as different as a regular lattice and hub-dominated network.

Regarding the specific point brought up by the reviewer, we would like to acknowledge that, indeed, it would be ideal to know a priori the optimal value of σ given a function of network properties. Actually, from our analysis, we have gained certain understanding of the properties that would be an argument of such a function (e.g., heterogeneity in the degree distribution, regularity of the network, etc.). However, that function seems to be far from being analytical, and we are not even in a position to provide an approximation.

Nevertheless, we want to emphasize two strengths of the methodology that make it applicable despite this shortcoming. First and more importantly, the recursive methodology to compute the numerical approximation of DomiRank is a very efficient and parallelizable algorithm, which allows us to explore the complete range of σ in affordable times, even for massive networks. The second point, and complementary to the previous one, is that the behavior of DomiRank does not vary chaotically with σ , exhibiting clear decreasing and increasing trends and even stable values for some topologies.

To clarify these points in our work, we have added text in the MS lines (500-514), as well as section S-III in the SM, where we display some illustrative examples of the loss functions for different topologies, including a massive network.

Speaking of σ - what were the values set in Figure 4?

The values have been reported in the revised version. Thanks!

Results. Finally, in some of the figures the advantage of DomiRank seems marginal (e.g., Fig. 3c, Fig. 4c,k...). This may be rectified to some extent if the authors add my suggested analysis of network topologies where DomiRank is most relevant. This will allow to showcase the "interesting" networks, where results will be significant. It will also help explain why, for example, they are not so impressive against the BA networks (which lack meso-scale structure, but have strong local heterogeneity). In any case, the authors may consider tuning down their language on how DomiRank outperforms all other methods, and acknowledge the cases where this "outperformance" is minor.

We thank the reviewer for bringing up this concern, as it has allowed us to clarify the point in the manuscript.

We want first to show our overall agreement with the general assessment of the reviewer that the results display different degrees of performance of DomiRank with respect to the other centralities, where in some specific cases, e.g. BA network is underwhelming (... still kind of impressive that DomiRank is able to mine the belittled global information - introduced by fine-size effects - amidst the dominating local information, to gain an edge over Degree and PageRank attacks in the particular topology!). That being said, the strength of DomiRank in revealing network fragility was not meant to be argued in the particular performance of any specific network but rather in its consistent competitive performance for all the topologies (both real and synthetic) tested. Thus, DomiRank, by modulating the intensity of the competition dynamics that underlies its definition, exhibits the ability to resonate with the network's fundamental properties to generate efficient attacks. For this reason, we decided to maintain all the panels in former Figure 4 (now Fig 5 in the amended manuscript) as its central message emerges from the joint results of all the panels rather than from the significance of the performance of DomiRank in any individual panel.

We have highlighted this message now in the MS - see lines (500-514).

We also believe that most of the changes incorporated in the amended manuscript (many nicely suggested by the reviewer) contribute as well to contextualizing the general message.

The paper's outline follows: 1. Presenting DomiRank, 2. Discussing its computational feasibility, 3. Showing results and discussing network insights. My feeling is that the potential readers will find item 3 much more important and interesting, and consider item 2 as a sideline discussion. If the authors agree with me on this, I recommend to reorganize the outline, first focusing on 3, then discussing item 2. Of course, its up to the authors to decide on that.

About this point, we must confess that we thought about this alternative outline when we wrote the manuscript. We have reconsidered once more, but we still believe that the current structure might be more logical as the low computational cost is a fundamental argument (and necessary condition) that supports the feasibility of the method despite the preprocessing needed to obtain the optimal value of the parameter. Thus, we hope that the reviewer agrees with our decision to keep the original outline.

As I indicated in the opening - I find this paper interesting, and think it can potentially lead to interesting generalizations down the road. I am looking forward to reconsidering a revised version.

We really appreciate the encouraging comments from the reviewer, and we want to thank the reviewer once more because their comments have been truly instrumental in improving the presentation of our work.

Small corrections:

Barabasi should appear with an accent on the second a (Barab^{\{s\}}si). See, e.g., lines 247 and 294.

Thanks! We have corrected it there and in other instances.

Line 410 "more directly", I suggest to replace with "directly". The qualifier "more" seems to be inappropriate here.

Done, thanks!

DomiRank is a ranking, hence any rescaling or gauging that preserves the rank order is permitted. Therefore, I suggest to decide whether Gamma is set to the range -1 to 1, or 0 to 1. There seems to be an inconsistency between Figs. 1 and 3.

We needed to use different scales depending on the purpose of the figure and, particularly if we were comparing among different centralities. In any case, now we have provided the details in each caption to avoid confusion.

Reviewer #3 (Remarks to the Author):

In this manuscript the authors introduce a new centrality measure, the DomiRank centrality, that exploits a competition mechanism among nodes, regulated by a free parameter σ . By opportunely tuning the free parameter, DomiRank centrality is able to modulate the effect of local vs global properties on the node relevance. The effectiveness in the dismantling of the network obtained with DomiRank-based attack is proven on a wide set of synthetic and real-world networks and it is shown that DomiRank-based attacks outperform other centralities-based attacks both in terms of the largest connected component and in terms of the fraction of links removed. The authors also show that DomiRank estimation allows for parallel computation and that the computational cost of recursively calculating DomiRank is relatively low.

The presented measure is of interest to the field and, overall, the study is exhaustive, the number of simulations and comparisons of the proposed quantity with other centrality measures is large enough to properly support DomiRank effectiveness.

We want to thank the reviewer for his summary and for supporting our results!

Nevertheless, I have some comments on the presented study that I think should be considered before publication:

- As with the DomiRank centrality, other centrality measures depend on some free parameter. For example, the Katz and PageRank centralities contain a free parameter whose value is chosen before the algorithm is applied and that must be set in the interval between 0 and the inverse of the largest (most positive) eigenvalue of A (in Katz centrality) or between 0 and the inverse of the largest eigenvalue of AD^{-1} , where D is the diagonal matrix with elements $D_{\{ii\}} = \max\left(k_i^{\text{out}}, 1\right)$ (for PageRank centrality). Do the authors explore the range of the free parameter for those centralities to find the optimal value to dismantle the network, as well as they explore the range of σ ? Otherwise, the comparison results would be biased and not properly conducted.

We thank the reviewer for this comment, as it allows us to clarify this point.

Indeed, in the results presented in the MS and SM, we have used only the default values of the parameters for Katz (0.01, or 0.001 when the centrality did not converge for 0.01 in 1000 iterations) and PageRank (0.85). However, the use of these parameters is not detrimental to the relative performance of those metrics with respect to the results shown in our work.

To provide evidence supporting the above claim and generally the methodology used in our work, we have added a new section in the SM (section S-III). This section includes a new figure, Fig. S4, reproduced here as Fig R11 to facilitate its discussion here. Fig R11 shows the loss (area under the LCC curve) as a function of the relative values of the parameters for Katz, PageRank, and DomiRank centrality for different network topologies consisting of $N = 300$ nodes, namely, Barabási-Albert, Erdős-Rényi, and Watts-Strogatz networks. From Fig R12, we observe that:

- i. The default parameters for Katz and PageRank are consistently in the area of the best performance for all the networks (despite their topological variability, except for Watts-Strogatz, where the loss is marginally worse). Very interestingly, this is not the case for DomiRank, wherein its optimal parameter varies for different topologies. This fact, far from being a weakness of the proposed centrality, is a strength, as it is a signature of the adaptiveness of DomiRank to mine different prevalent topological properties at various ranges of its parameter.
- ii. The curves are relatively flat around the optimal value, meaning there are small differences in using the truly optimal parameter for PageRank and Katz and their default value.

Given the above arguments, using the default parameter values for PageRank and Katz is preferred, as it does not significantly affect the results or alter the message. This choice has been clearly stated in the amended version of the manuscript in lines 391-394.

Fig R11 – Comparing the loss functions of PageRank, Katz, and DomiRank within their parameter space. The loss functions corresponding to PageRank, Katz, and DomiRank computed for the complete range of their free parameters are shown for three different synthetic networks of size $N = 300$, namely, (a) Barabási-Albert, (b) Erdős-Rényi, and (c) Watts-Strogatz. Note that all the curves shown were computed using 25 samples linearly distributed in their respective parameter domains. *Note that this figure has been included in the SM as Fig. S4.*

- More insights into the analytical formulation of DomiRank centrality should be provided. How is it related to other measures like Katz or PageRank? Are the values of centrality measured at the limit $\sigma \rightarrow \frac{1}{\lambda_N}$ similar to the ones measured with other methods? Which, if any, pitfalls of existing centrality measures is DomiRank able to solve?

We thank the reviewer for these comments, which are in line with similar comments brought up by another reviewer. As the reviewer will be able to examine, we have extensively modified that part of the MS to provide more insight into the analytical expression and the implications of the competition/dominance dynamics and the resulting scores. Particularly, the reviewer will note that a significant effort has been made in re-writing section II, where DomiRank is introduced. This section has been now divided into subsections, where:

- Subsection A presents the definition and interpretation of DomiRank, including now the nodal formulation of Equation 1 to gain interpretability of the modes.
- Subsection B presents a detailed discussion of the role of DomiRank's parameter (σ). Within this subsection, Figure 1 has been modified, along with the addition of Figure 2, highlighting how and why DomiRank converges to degree distribution as σ tends to zero.
- Subsection C provides the rationale behind the ability of DomiRank to reveal network fragility, which we believe is instrumental in contextualizing DomiRank with respect to the different centralities.

Also, former Figure 3 (now Figure 4 in the amended manuscript) has been modified to include the visualization of different stages of network dismantling during the attack sequences. Those instances were selected to highlight fundamental differences in the assessments made by the different centralities. A detailed discussion can be found now in lines 397-437 in the new version of our manuscript.

To bring further light to the point brought up by the reviewer regarding the similarities and differences between DomiRank and other centralities, we have added a new section in the SM (S-IV), where the correlations between all the analyzed metrics are explored and represented there in Figure S5, which is reproduced here as Fig R12 for discussion purposes. Fig. R12 shows the expected progressive divergence of DomiRank from degree centrality. We also observe that DomiRank displays different degrees of correlation with different centralities depending on the topology examined, providing further evidence of the adaptability of DomiRank by varying the value of σ to mine the most relevant structural features of each topology. Note that a more detailed discussion has been included in section S-IV in the new version of the SM.

a) Barabási - Albert $\sigma = \frac{0.13}{-\lambda_N}$

b) Erdős - Rényi $\sigma = \frac{0.77}{-\lambda_N}$

c) Random Geometric $\sigma = \frac{0.87}{-\lambda_N}$

d) 2D - Lattice $\sigma = \frac{0.999}{-\lambda_N}$

Fig R12 - Correlation between centralities. Pairwise Spearman correlation matrices for 11 centralities are presented as heatmaps for four different topologies, namely, (a) Barabási–Albert ($N = 1000$), (b) Erdős–Rényi ($N = 1000$), (c) Random Geometric Graph ($N = 1000$), and a regular 2D-Lattice ($N = 33 \times 33$). Particularly, the centralities compared are Eigenvector (EV), Harmonic (HM), Closeness (CN), Betweenness (BN), Load (LO), Current-Flow (CF), Katz (KA), Degree (DG), PageRank (PR), DomiRank (DR), and Collective Influence (CI). *Note that this figure has been included in the SM as Fig. S5.*

- How is DomiRank analytically defined in directed networks? That is, how is it defined in asymmetric adjacency matrices? The authors introduce the case of directed graphs in section III A, a section related to the evaluation of the proposed measure, whereas a formal definition should be provided in section II.

We thank the reviewer for this comment! Indeed, DomiRank is defined for both directed and undirected networks. We have clarified this in the main text, both in lines 113-117 and lines 190-197.

Another specific remark that would improve paper readability is the following:

- In Fig 3, the evaluated optimal values of σ for the different network types are reported. From those quantities, the reader has a hint on how optimal parameter values are related to different meso- and macroscale structural features. To this aim, the selected σ could be included also in the other figures, for example in Fig 4.

Great suggestion! We have added those values and thank the reviewer for pointing out this.

REVIEWERS' COMMENTS

Reviewer #1 (Remarks to the Author):

The authors have successfully and comprehensively addressed all of my concerns. I commend the authors on an excellent revision and a fine study overall. I recommend publication with no further comments or requests.

Very best,

Douglas Guilbeault

Haas School of Business

University of California, Berkeley

Reviewer #2 (Remarks to the Author):

I think the authors have done a great job at improving their paper and addressing my comments. In fact, having now read the entire response file - I think the discussion itself, and the authors' detailed and illustrated response should itself be published, as a good example of to write a "response to referees". This is of course, up to the authors/editors/other Referees - so I leave it just as a suggestion.

Bottom line - I now recommend publication without further revisions.

REVIEWER COMMENTS

Reviewer #1 (Remarks to the Author):

The authors have successfully and comprehensively addressed all of my concerns. I commend the authors on an excellent revision and a fine study overall. I recommend publication with no further comments or requests.

Very best,
Douglas Guilbeault
Haas School of Business
University of California, Berkeley

We want to thank the reviewer for his kind words. His feedback during the review process has been particularly inspiring and enjoyable, and was instrumental in improving the readability and scope of our paper.

Reviewer #2 (Remarks to the Author):

I think the authors have done a great job at improving their paper and addressing my comments. In fact, having now read the entire response file - I think the discussion itself, and the authors' detailed and illustrated response should itself be published, as a good example of to write a "response to referees". This is of course, up to the authors/editors/other Referees - so I leave it just as a suggestion.

Bottom line - I now recommend publication without further revisions.

All my best,
Baruch Barzel

We thank the reviewer for his compliments on our work. Regarding his comment about publishing the 'response to referees' as a good example, we must say that we are flattered by this comment and if the editor also thinks that this could be valuable, definitely we would be fully supportive as well. Nevertheless, we want to point out that during the initial submission process we selected the option to make available the 'peer review file' in case of acceptance, so therefore this file will at least be publicly available.

We would also like to take this opportunity to thank the reviewer again for his stimulating comments, which significantly contributed to a clearer and more insightful manuscript.